# Dysregulated heparan sulfate proteoglycan metabolism promotes Ewing sarcoma tumor growth

Elena Vasileva[1], Mikako Warren[2,3], Timothy J Triche[2,3], James F Amatruda[1,4,5]*

[1]Cancer and Blood Disease Institute, Children's Hospital Los Angeles, Los Angeles, United States; [2]Division of Pathology and Laboratory Medicine, Children's Hospital Los Angeles, Los Angeles, United States; [3]Department of Pathology, Keck School of Medicine, University of Southern California, Los Angeles, United States; [4]Department of Pediatrics, Keck School of Medicine, University of Southern California, Los Angeles, United States; [5]Department of Medicine, Keck School of Medicine, University of Southern California, Los Angeles, United States

*For correspondence:
jamatruda@chla.usc.edu

Competing interest: The authors declare that no competing interests exist.

**Abstract** The Ewing sarcoma family of tumors is a group of malignant small round blue cell tumors (SRBCTs) that affect children, adolescents, and young adults. The tumors are characterized by reciprocal chromosomal translocations that generate chimeric fusion oncogenes, the most common of which is EWSR1-FLI1. Survival is extremely poor for patients with metastatic or relapsed disease, and no molecularly targeted therapy for this disease currently exists. The absence of a reliable genetic animal model of Ewing sarcoma has impaired investigation of tumor cell/microenvironmental interactions in vivo. We have developed a new genetic model of Ewing sarcoma based on Cre-inducible expression of human *EWSR1-FLI1* in wild-type zebrafish, which causes rapid onset of SRBCTs at high penetrance. The tumors express canonical EWSR1-FLI1 target genes and stain for known Ewing sarcoma markers including CD99. Growth of tumors is associated with activation of the MAPK/ERK pathway, which we link to dysregulated extracellular matrix metabolism in general and heparan sulfate proteoglycan catabolism in particular. Targeting heparan sulfate proteoglycans with the specific heparan sulfate antagonist Surfen reduces ERK1/2 signaling and decreases tumorigenicity of Ewing sarcoma cells in vitro and in vivo. These results highlight the important role of the extracellular matrix in Ewing sarcoma tumor growth and the potential of agents targeting proteoglycan metabolism as novel therapies for this disease.

## Editor's evaluation

This model represents an improvement upon previous zebrafish sarcoma models and the data suggest that the methods employed yield tumors that resemble human disease. This new model may be used to better understand sarcoma progression so that new therapeutic targets may be realized.

## Introduction

Ewing sarcoma is an aggressive sarcoma of bone and soft tissue with a peak incidence in adolescents and young adults (*Gaspar et al., 2015*). Diagnosis relies on histologic and molecular analysis of biopsy specimens or surgically resected tumor tissue. Histologic examination of the tumor typically reveals sheets of small, round, blue cells with a prominent nucleus and scant cytoplasm. Approximately 20–25% of patients present with metastases at diagnosis that are often resistant to intensive

therapy (*Gaspar et al., 2015*). Standard multimodal therapy for patients with small round blue cell tumor (SRBCT) includes surgical resection and/or local radiotherapy as well as intensive multiagent chemotherapy (*Grünewald et al., 2018*). Ewing sarcoma family tumors are characterized by the presence of reciprocal chromosomal translocations that fuse a member of the FET family of RNA-binding proteins (encoded by FUS, EWSR1, and TAF15), with different members of the ETS (E26-specific) family of transcription factors. The most common oncofusion found in 85% of cases is *EWSR1-FLI1* (*Delattre et al., 1992*; *Grünewald et al., 2018*). These chimeric fusion oncoproteins act as aberrant transcription factors deregulating hundreds of genes (e.g., genes involved in cell-cycle regulation, cell migration, and proliferation) by binding DNA enriched with GGAA motifs (*Gangwal et al., 2008*; *Guillon et al., 2009*; *Johnson et al., 2017*). These and other studies, made on patient-derived tumor tissue and cell lines, have provided great insight into the molecular mechanisms of fusion protein function. However, in the absence of a representative in vivo model, other questions, including the cell of origin of Ewing sarcoma, mechanisms of tumor initiation, and the relationships between tumor and host cells, remain a subject of constant debate.

ERK1 and ERK2 are key effectors in the Ras–Raf–MEK–ERK signal transduction cascade. The phosphorylation of ERK1/2 is required for the activation and subsequent phosphorylation of hundreds of cytoplasmic and nuclear substrates including transcription factors regulating cell adhesion, migration, and proliferation (*Meloche and Pouysségur, 2007*). ERK1/2 activity is required for transformation of NIH-3T3 cells by EWSR1-FLI1 (*Silvany et al., 2000*). In therapy-naive Ewing sarcoma samples, combined expression of pAKT, pmTOR, and pERK predicted worse progression-free and overall survival (*van de Luijtgaarden et al., 2013*). ERK is a downstream target of insulin-like growth factor signaling, however the prognostic impact of IGF-1 and IGF-1 receptor expression in Ewing sarcoma is controversial (*Scotlandi et al., 2011*). These findings raise the question of precisely which mechanisms drive ERK1/2 signaling in Ewing sarcoma, including the possible role of the tumor microenvironment.

The tumor microenvironment plays an important role in determining tumor cell growth, survival, and response to treatment. The tumor microenvironment is composed of various cell types embedded in an altered extracellular matrix (ECM). The ECM is a three-dimensional network of extracellular proteins, collagen, glycoproteins, and signaling molecules that provide structural and biochemical support to surrounding cells, as well as playing an essential role in signal transduction (*Kim et al., 2011*). Proteoglycans are key molecular effectors of the cell surface and pericellular microenvironment with essential roles in signal transduction and regulating cell adhesion, migration, and differentiation. Signaling between cells is modulated by proteoglycan activity at the cell membrane (*Elfenbein and Simons, 2010*). They have an ability to interact with both ligands and receptors that could directly affect cancer growth (*Edwards, 2012*; *Iozzo and Sanderson, 2011*; *Multhaupt et al., 2016*; *Mythreye and Blobe, 2009*). Importantly, heparan sulfate proteoglycans play an essential role in differentiation and migration of both neural crest and mesenchymal cells – both of them are considered as potential cell of origin of Ewing sarcoma (*Henderson and Copp, 1997*; *Long and Huttner, 2019*; *Papy-Garcia and Albanese, 2017*; *Yaylaci et al., 2016*). To date, however, little is known about the role of proteoglycans and activation of ERK1/2 signaling in Ewing sarcoma development.

A genetic animal model of Ewing sarcoma family of tumors would thus be highly valuable as a complement to existing xenograft models in both mouse and zebrafish (*El-Naggar et al., 2015*; *Grünewald et al., 2018*) for exploration of cooperating genetic factors for tumor development and for testing the role of the complex microenvironment in tumor progression. However, multiple attempts to create genetically engineered mouse models of Ewing sarcoma have not yielded a tractable model, likely due to the developmental toxicity of heterotopic expression of the oncofusion (*Minas et al., 2017*). Previously, we demonstrated that transposon-mediated expression of human *EWSR1-FLI1* drives SRBCT formation in zebrafish from 6 to 19 months of age (*Leacock et al., 2012*). While serving as proof of principle that *EWSR1-FLI1* is tumorigenic in fish, the model had some limitations, including low penetrance, requirement for tp53 deficiency, and a low incidence of other tumor types such as leukemias.

Here, we describe a Cre-inducible invasive model of Ewing Sarcoma in zebrafish that reproducibly develops tumors in wild-type backgrounds. We characterize tumors and show that they recapitulate the main aspects of the human disease. Using this model, we show that tumor growth is associated with the activation of the ERK1/2 signaling pathway, and that EWSR1-FLI1 upregulates expression of proteins involved in extracellular matrix reorganization and heparan sulfate proteoglycan catabolism.

We demonstrate that surfen, a heparan sulfate antagonist, effectively reduces proteoglycan-mediated activation of ERK1/2 signaling in Ewing sarcoma cell lines and zebrafish models, leading to decreased proliferation of Ewing sarcoma tumor cells in vitro and in vivo.

## Results

### Cre-inducible expression of *EWSR1-FLI1* drives SRBCT development in zebrafish

We reasoned that the low penetrance of tumors and requirement for loss of tp53 function in our original zebrafish Ewing sarcoma model (*Leacock et al., 2012*) might be due to developmental toxicity of *EWSR1-FLI1*, similar to what was described in mouse models (*Minas et al., 2017*). This is especially true since microinjection is performed into single-cell-stage embryos, allowing integration of transposons into cell populations early in development. We therefore tested several strategies, beginning with expression of *EWSR1-FLI1* under ubiquitous and tissue-specific promoters in wild-type zebrafish embryos (*Figure 1A*). All constructs incorporated eGFP (Enhanced Green Fluorescent Protein) linked to *EWSR1-FLI1* via a viral 2A linkage, to allow fluorescent labeling of *EWSR1-FLI1*-expressing cells via an unfused eGFP (*Provost et al., 2007*). Consistent with our previous data (*Leacock et al., 2012*), constitutive expression of eGFP-2A-*EWSR1-FLI1* under the *beta-actin* promoter in wild-type fish caused high embryonic lethality and low incidence of tumor formation (*Figure 1—figure supplement 1A*). We obtained similar results with other ubiquitous promoters including *cmv* and ubiquitin (*ubi*). On the other hand, expression of *EWSR1-FLI1* from tissue-specific promoters (s*ox9b*, *fli1a*, and *mitfa*) failed to produce tumors (*Figure 1—figure supplement 1A*, data not presented). Some of these promoters become active only during somitogenesis or at later stages, suggesting that the developmental stages or lineages marked by these promoters were no longer susceptible to transformation by *EWSR1-FLI1*.

Based on these results, we suspected that at early stages of development, beyond the initial pregastrulation stage but prior to the onset of somitogenesis, there might be a population of cells that would be susceptible to transformation. We accordingly designed a Cre-inducible allele (loxP-DsRed-STOP-loxP-eGFP-2A-*EWSR1-FLI1*, henceforth Red-STOP-Green or '*RSG*'-*EWSR1-FLI1*) that would allow more control over the timing of *EWSR1-FLI1* expression. Injection of *RSG-EWSR1-FLI1* transposon and *Tol2* transposase, along with mRNA encoding Cre, allows for a short delay in expression of *EWSR1-FLI1*, as the *Cre* mRNA must be translated before recombination can occur. Initial attempts to generate tumors with *RSG-EWSR1-FLI1* driven by the beta-actin promoter (*Figure 1—figure supplement 1B* and *Figure 1—figure supplement 2B, C*) caused early apoptosis (*Figure 1—figure supplement 2A*) and the growth of cranial cell masses (*Figure 1—figure supplement 2D*), however failed to recapitulate the histologic appearance of Ewing sarcoma (*Figure 1—figure supplement 2E*).

In contrast, coinjections of *Cre* mRNA with expression constructs in which *RSG-EWSR1-FLI1* is driven by the ubiquitin promoter led to robust, mosaic expression of *EWSR1-FLI1* and e*GFP* (*Figure 1A, B*). To estimate the frequency of tumor development using this approach, injected fish were sorted for eGFP at 14 days post-fertilization (dpf) and a cohort of 77 fish was monitored up to 4 months. To control for promoter leakiness, we injected embryos with *ubi:RSG-EWSR1-FLI1* in the presence of GFP RNA (142 fish). Uninjected embryos were used as an additional negative control (150 fish). We found that 34% of eGFP-positive fish developed tumors with onset during the first 3–4 weeks post-fertilization (*Figure 1C*) exhibiting SRBCT histology (*Figure 1D*, *Figure 1—figure supplement 3C*). No tumors were observed in either control group. Among the fish developing tumors, 42% developed a single tumor, while 58% of fish had two or more tumors (*Figure 1—figure supplement 3A*). *EWSR1-FLI1*-driven tumors arose most frequently in the dorsal regions of the fish (*Figure 1E*). Thirty-seven percent of tumors were SRBCTs associated with the dorsal fin radial bones, while a further 15% of tumors arising from similar locations showed deep muscle invasiveness and were characterized as SRBCT with diffuse striated muscle infiltration. 9% and 21% of tumors were associated with the skeleton of caudal and anal fins, respectively. Finally, we observed several tumors that appeared to arise from the cranial skeleton or the brain. In summary, the inducible *ubi:dsRed-stop-eGFP2A-EWSR1-FLI1* allele efficiently induced bone and soft-tissue small round blue cell sarcomas in fish, without requirement for impaired tp53 function.

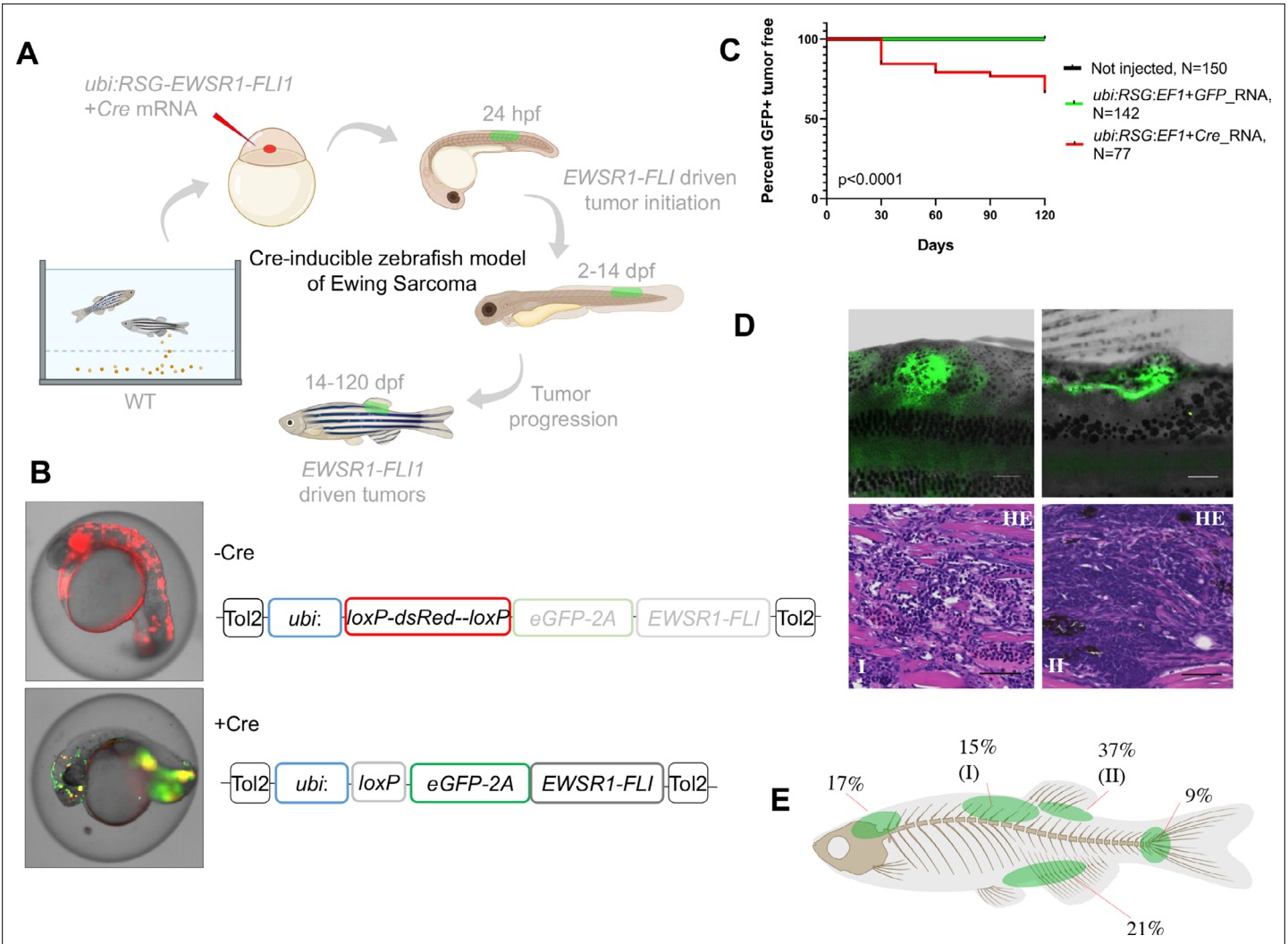

**Figure 1.** Cre-inducible expression of *EWSR1-FLI1* drives small round blue cell tumor (SRBCT) development in zebrafish. (**A**) Overview of experimental method. The Tol2 transposon system was used to integrate human *EWSR1-FLI1* into the wild-type zebrafish genome by microinjection into single-cell-stage embryos. (enhanced Green Fluorescent Protein) eGFP-positive fish were monitored up to 4 months. (**B**) Constructs for Cre-inducible expression of *EWSR1-FLI1*. (**C**) Incidence of eGFP+ tumors detected in *Ubi:RSG-EWSR1-FLI1; Cre* RNA (*n* = 77) injected zebrafish versus *Ubi:RSG-EWSR1-FLI1; eGFP* RNA (*n* = 142) and uninjected controls (*N* = 150) in a wild-type genetic background (p < 0.001 by log-rank test). (**D**) Representative images of eGFP-positive tumors in zebrafish (top panel) and H&E staining of tumor sections (bottom panel): (I) SRBCT with diffuse skeletal muscle infiltration and (II) visceral SRBCT arising from fin dorsal radial bone. Scale bars, 100 μm. (**E**) Percent tumor incidence at different anatomic sites.

The online version of this article includes the following figure supplement(s) for figure 1:

**Figure supplement 1.** The Tol2 transposon-based system was used to integrate *EWSR1-FLI1* into the zebrafish genome by microinjection into single-cell-stage zebrafish embryos.

**Figure supplement 2.** The Tol2 transposon-based system was used to integrate *b-actin:RSG-EWSR1-FLI1* into the wild-type zebrafish genome.

**Figure supplement 3.** Mosaic model of Ewing sarcoma generated by the microinjection of *ubi:RSG-EWSR1-FLI1* plus *Cre* mRNA into single-cell-stage embryos.

## *EWSR1-FLI1*-driven tumors in zebrafish recapitulate the main aspects of human Ewing sarcoma

To further characterize the zebrafish model, we performed immunohistochemistry (IHC) on tumors with antibodies specific for human FLI1 and CD99 (*Figure 2A*, *Figure 2—figure supplement 1A, C*). As expected, tumor cells had nuclear localization of EWSR1-FLI1, whereas control fish had no expression of FLI1 in the corresponding sites (*Figure 2—figure supplement 1A*, top panel). CD99 is a cell surface glycoprotein that serves as a sensitive, clinically useful marker for Ewing sarcoma (*Muhammad*

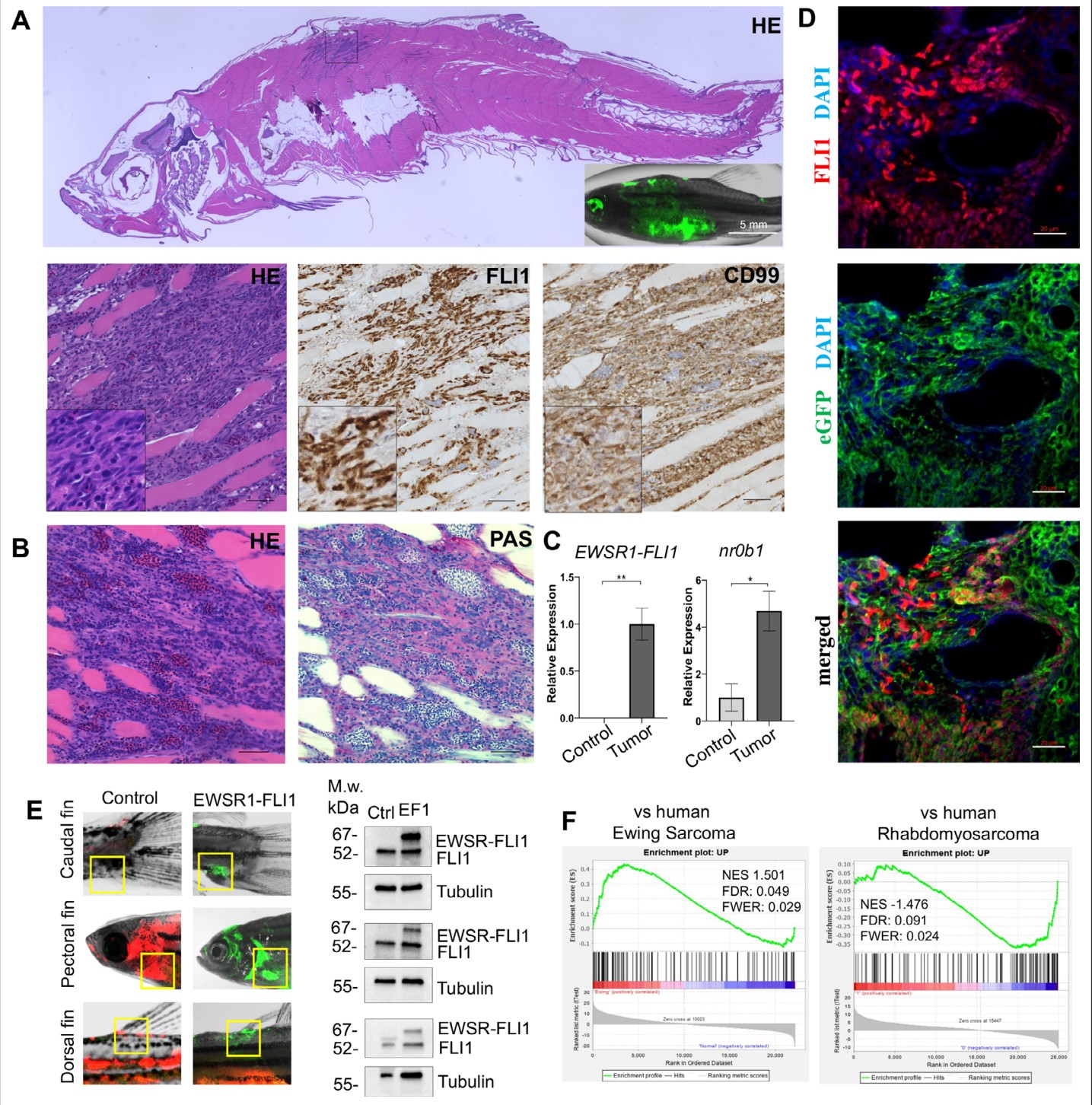

**Figure 2.** Zebrafish tumors phenocopy human Ewing sarcoma. (**A**) Representative image of Ewing sarcoma tumor in WT zebrafish. H&E staining, immunohistochemistry (IHC) staining with anti-FLI1 and anti-CD99 antibodies. Scale bars, 100 μm. (**B**) H&E and Periodic acid-Schiff (PAS) staining of zebrafish tumors. Scale bars, 50 μm. (**C**) Relative mRNA expression of human *EWSR1-FLI* and zebrafish *nr0b1* in normal and tumor tissues. Error bars represent standard error of the mean (SEM), $N = 3$, *$p < 0.05$, **$p < 0.01$, two-tailed Student's *t*-test. (**D**) Immunofluorescence staining of zebrafish tumor with anti-eGFP and anti-FLI1 antibodies. Scale bars, 20 μm. (**E**) Validation of *EWSR1-FLI1* expression in tumors at different sites by immunoblotting. (**F**) Gene set enrichment analysis (GSEA) comparing the enrichment of common upregulated proteins at dorsal, caudal and pectoral fin tumors to human Ewing sarcoma (GSE17674) and human Rhabdomyosarcoma (GSE108022) datasets.

The online version of this article includes the following figure supplement(s) for figure 2:

**Figure supplement 1.** Validation of zebrafish model of Ewing sarcoma.

et al., 2012). IHC confirmed the presence of CD99 on the cell surface of zebrafish Ewing sarcoma tumor cells (*Figure 2A*, *Figure 2—figure supplement 1C*).

Ewing sarcoma cells are characterized by the presence of the glycogen and are positive for Periodic acid-Schiff (PAS) staining (*Muhammad et al., 2012*). Consistently, zebrafish tumors were positive for PAS staining (*Figure 2B*). *NR0B1* is a target of EWSR1-FLI1 that directly modulates transcription and oncogenesis in Ewing sarcoma (*Kinsey et al., 2009*). Analysis of relative RNA expression showed upregulated expression of *nr0b1* in tumor tissues compared to normal (*Figure 2C*), correlating with *EWSR1-FLI1* expression level in tumor samples (*N* of repeats = 3, *N* of replicas = 3). Recently, it was shown that Ewing sarcoma cells exhibit cell-to-cell heterogeneity affecting proliferative and migratory potential of tumor cells (*Franzetti et al., 2017*). To test whether zebrafish tumor cells express *EWSR1-FLI1* on different levels we performed immunofluorescence staining of tumor sections with anti-eGFP and anti-FLI1 antibodies (*Figure 2D*). eGFP staining was used to label all tumor cells within the tumor, while FLI1 staining was implemented to detect the EWSR1-FLI oncofusion in tumor cells (*Figure 2D*). Staining of zebrafish tumors showed that cells have varying levels of EWSR1-FLI1. Thus, the zebrafish model appears to recapitulate the cell-to-cell heterogeneity found in human Ewing sarcoma cells.

To test whether tumors express EWSR1-FLI1 on the protein level we collected tumor and normal tissues at different sites (dorsal, caudal, and pectoral fins) and processed them for immunoblot analysis. All tumor samples expressed EWSR1-FLI1 (*Figure 2E*) (*N* of repeats = 3, *N* of replicas = 3). We next performed liquid chromatography–mass spectrometry (LC–MS/MS) analysis of protein samples made from single tumor dissected from dorsal, caudal, and pectoral fins. Normal tissues from the corresponding sites were used as controls. We first identified a set of 194 proteins that were commonly upregulated in the zebrafish tumors compared to control tissues (p < 0.05). To test the similarity of fish and human Ewing sarcoma, we used gene set enrichment analysis (GSEA) (*Subramanian et al., 2005*) to test the enrichment of the 194 upregulated proteins in a publicly available RNA-seq dataset of 44 human Ewing sarcoma tumors and 18 normal samples Series (GSE17674). Previously, it was shown that differentially expressed mRNAs correlate significantly with their protein products (*Koussounadis et al., 2015*). As a further test of specificity, we compared the fish tumors to another RNA-seq dataset of 44 human rhabdomyosarcoma tumors and 5 normal samples (GSE108022). GSEA showed enrichment of zebrafish tumor upregulated genes (NES = 1.501, FWER = 0.029) in the human Ewing sarcoma dataset but not in human rhabdomyosarcoma (NES = −1.476, FWER = 0.024) (*Figure 2F*). To enhance our analysis, we used publicly available data of proteins significantly upregulated in human mesenchymal cells after the expression of EWSR1-FLI1, considering that expression of the oncofusion in normal human cells best mimics the transformation process (*Tanabe et al., 2018*) (oncotarget-09-14428 s009). We used that list of upregulated proteins to run the GSEA to evaluate the enrichment of those hits in our zebrafish tumors dataset (*Figure 2—figure supplement 1B*). We found that the list of proteins was significantly upregulated in zebrafish tumors (NES: 1.540, FWER p value: 0.048) confirming similarity between zebrafish and human disease (*Figure 2—figure supplement 1B*). Taken together, the histologic appearance, marker expression, and gene expression establish the similarity of zebrafish EWSR1-FLI1-induced sarcomas to human Ewing sarcoma.

## Characterization of embryonic model of ES

In most murine models, *EWSR1-FLI1* expression is toxic and causes embryonic lethality (*Minas et al., 2017*). To estimate the developmental toxicity of *EWSR1-FLI1* expression in zebrafish we performed survival analysis in a group of 524 animals. Uninjected embryos (*N* = 528), embryos injected with *ubi:RSG-EWSR1-FLI1* and *GFP* mRNA (*N* = 328) and embryos injected with *ubi:RSG* and *Cre* mRNA (*N* = 304) were used as negative controls. Expression of *EWSR1-FLI1* significantly increased embryonic mortality in zebrafish (*Figure 3A*).

To characterize the effects of *EWSR1-FLI1* expression during early stages of zebrafish development we integrated the *ubi:RSG-EWSR1-FLI1* cassette into the zebrafish genome in the presence of *Cre* mRNA (*Figure 3B*, top panel). Negative controls included coinjection of *Cre* mRNA with an *ubi:RSG* construct that lacks *EWSR1-FLI1* (*Figure 3B*, middle panel) and coinjection of *ubi:RSG-EWSR1-FLI* with *GFP* mRNA (*Figure 3B*, bottom panel). Embryos were imaged at 12 hr post-fertilization (hpf), 24 hpf, and 5 dpf. The timeline demonstrates that eGFP driven by the *ubi* promoter is expressed broadly, including throughout the muscle (*Figure 3B*, middle panel). However, zebrafish expressing

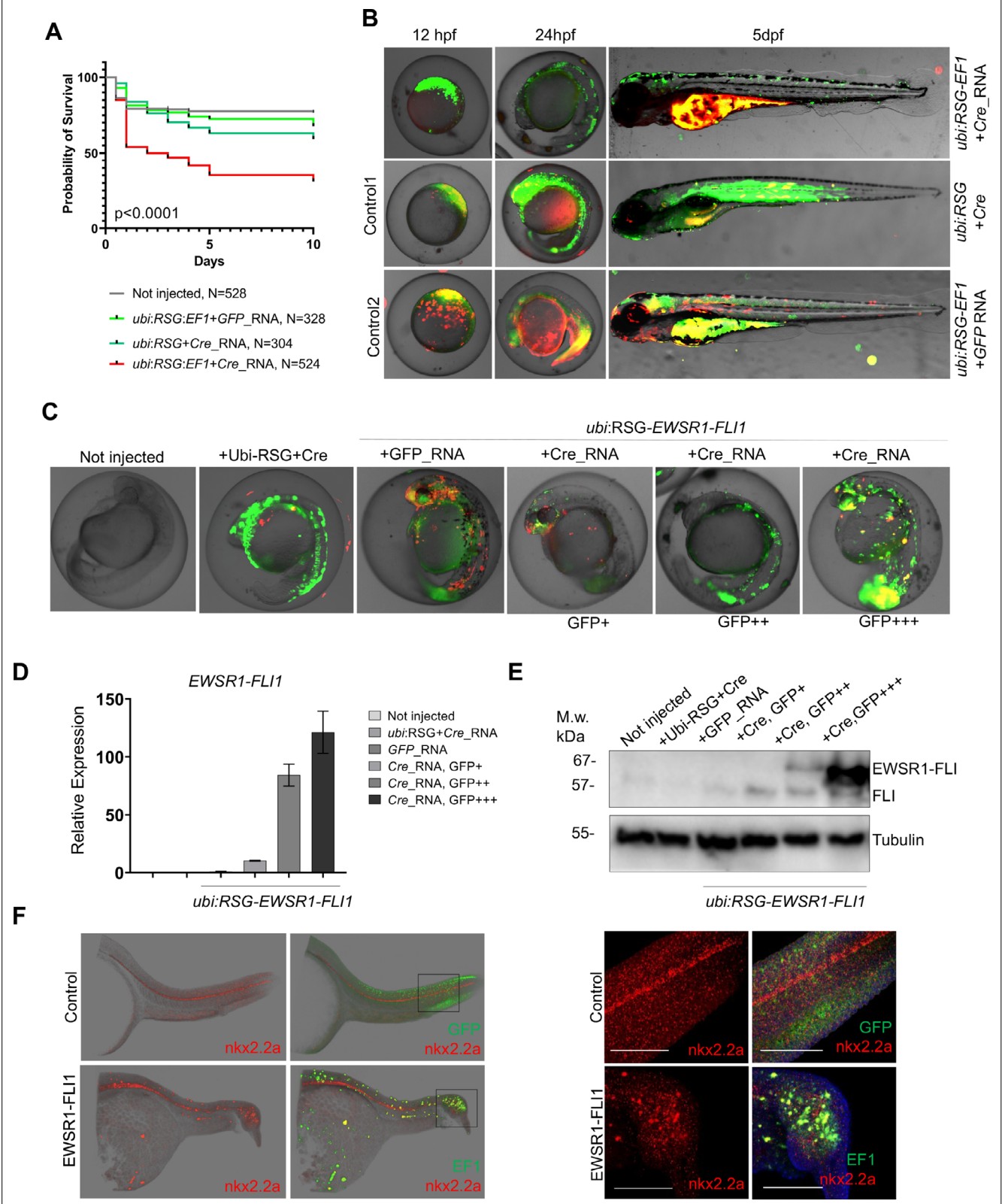

**Figure 3.** Validation of *EWSR1-FLI* expression in zebrafish embryos. (**A**) Rate of embryos expressing *EWSR1-FLI1* during the first 10 days of development (*N* = 524 Survival). Uninjected embryos (*N* = 528) as well as embryos injected with *ubi*:RSG-*EWSR1-FLI1* plus *GFP* mRNA (*N* = 328) or *ubi*:RSG plus *Cre* mRNA (*N* = 304) were used as negative controls (p < 0.001 by log-rank test). (**B**) Timeline of zebrafish development after injection with *ubi*:RSG-*EWSR1-FLI1* plus *Cre* mRNA (top panel). Zebrafish injected with *ubi*:RSG-*EWSR1-FLI1* plus *GFP* mRNA (middle panel) or *ubi*:RSG plus *Cre* mRNA (bottom panel)

*Figure 3 continued on next page*

*Figure 3 continued*

were used as negative controls. Images were taken at 12 hpf, 24 hpf, and 5 dpf time points. (**C**) Representative image of embryos with low (GFP+), medium (GFP++), and high (GFP+++) levels of *EWSR1-FLI1* expression. (**D**) Relative mRNA expression of *EWSR1-FLI1* in embryos with low (GFP+), medium (GFP++), and high (GFP+++) levels of *EWSR1-FLI1* according to eGFP signal. (**E**) Immunoblot confirming the expression of EWSR1-FLI1 protein in embryos with low (GFP+), medium (GFP++), and high (GFP+++) levels of EWSR1-FLI1 expression according to eGFP signal. (**F**) RNA scope staining of control and EWSR1-FLI1-positive embryos for nkx2.2a (529751-C2 RNAscope Probe – Dr-nkx2.2a-C2) and eGFP (538851 RNAscope Probe – EGFP-O4).

eGFP-2A-EWSR1-FLI1 under the *ubi* promoter have a distinct distribution pattern of eGFP-positive cells (*Figure 3B*, upper panel), predominantly on the fish dorsum, tail, and fins (*Figure 3B*, upper panel).

To confirm the expression of *EWSR1-FLI1* in the zebrafish embryo model, we collected embryos expressing low (GFP+), medium (GFP++), and high (GFP+++) levels of *EWSR1-FLI1* (*Figure 3C*) for analysis via qRT-PCR (*Figure 3D*) or immunoblot (*Figure 3E*). Each sample consisted of 10 embryos at 24 hpf (*N* of repeats = 3, *N* of replicas = 3). Upon Cre-mediated recombination, *EWSR1-FLI* is efficiently expressed in embryos on both the RNA and protein levels (*Figure 3D, E*).

We also found that EWSR1-FLI1 drives the formation of outgrowths in embryonic model of the disease (*Figure 3F*). To better characterize the EWSR1-FLI1-expressing cells we evaluated *nkx2.2a* gene expression in zebrafish outgrowths. NKX2.2 is an immunohistochemical marker that has been reported to be sensitive and specific for Ewing sarcoma in human (*McCuiston and Bishop, 2018*). Because there are no available antibodies for zebrafish nkx2.2a we optimized an RNA scope approach to evaluate RNA expression of *nkx2.2a* in whole mount embryos. We generated fish mosaically expressing *EWSR1-FLI1* and sorted animals with *EWSR1-FLI1*-induced tumors at 24 hpf. Embryos were fixed in 4% paraformaldehyde (PFA and used for double RNAscope staining for *nkx2.2a* (529751-C2 RNAscope Probe – Dr-nkx2.2a-C2)) and eGFP (538851 RNAscope Probe – EGFP-O4). Normally *nkx2.2a* is expressed in the spinal cord (*Figure 3F*, control). We found that consistent with human data the *EWSR1-FLI1*-expressing cells in tumor were also positive for *nkx2.2a* (*Figure 3F*, EWSR1-FLI).

In summary, the *Cre*-inducible allele leads to robust and reproducible expression of human *EWSR1-FLI1* with limited spatial distribution during early zebrafish embryogenesis. Developmental toxicity, while much less than that caused by *EWSR1-FLI1* expression under *b-actin* and *cmv* promoters, does impact the survival of larvae over the first 10 days of life. Finally, EWSR1-FLI1 drives the formation of *nkx2.2a*-positive outgrowths in embryonic model of the disease.

## EWSR1-FLI1 expression leads to activation of ERK1/2 signaling in zebrafish embryos and adult tumors

Our finding that expression of *EWSR1-FLI1* in early embryos leads to a high prevalence of tumors in adult zebrafish led us to examine the behavior of cells expressing *EWSR1-FLI1* in developing embryos, to understand early events in tumorigenesis. Zebrafish embryos mosaically expressing *EWSR1-FLI1*, but not control embryos, developed outgrowths visible as discrete cell masses. Immunostaining with anti-phospho-histone H3 (pH3) showed increased cell proliferation in these areas, associated with expression of *EWSR1-FLI1* as indicated by the associated eGFP marker (*Figure 4A*). The Ras–MAPK–ERK signaling pathway transduces signals downstream of growth factor receptors, and is an important mediator of cell proliferation during embryonic development and in cancer (*Kamiya et al., 2015*; *Maekawa et al., 2007*; *Steinmetz et al., 2004*; *Wong et al., 2018*; *Yang et al., 2020*; *Zhou et al., 2019*). To assess the contribution of MAPK–ERK signaling to formation of outgrowths, we performed immunofluorescence staining for the active, phosphorylated form of ERK1/2 (pERK1/2). Cells expressing *EWSR1-FLI1* were positive for pERK1/2 (*Figure 4B*). While most pERK1/2-positive cells also expressed *EWSR1-FLI1* (*Figure 4C*, *Videos 1 and 2*), some surrounding cells lacking the transgene were also marked with pH3 and pERK1/2 (*Figure 4—figure supplement 1A*). Thus, activation of ERK1/2 signaling is an early event in *EWSR1-FLI1*-driven aberrations.

To test whether the activation of ERK1/2 signaling is also a feature of mature zebrafish tumors we performed ERK1/2 pathway analysis in zebrafish tumors. We used GSEA to determine the enrichment of genes associated with ERK1/2 pathway (GOBP_ERK1_AND_ERK2_CASCADE) in zebrafish tumors. We found that the GOBP_ERK1_AND_ERK2_CASCADE gene set was significantly enriched in zebrafish tumor proteomic dataset (NES:1.3867193, FWER p value: 0.03) (*Figure 4D*). To confirm this finding, we prepared histologic sections of tumors and performed IHC for pERK1/2 (*Figure 4E*,

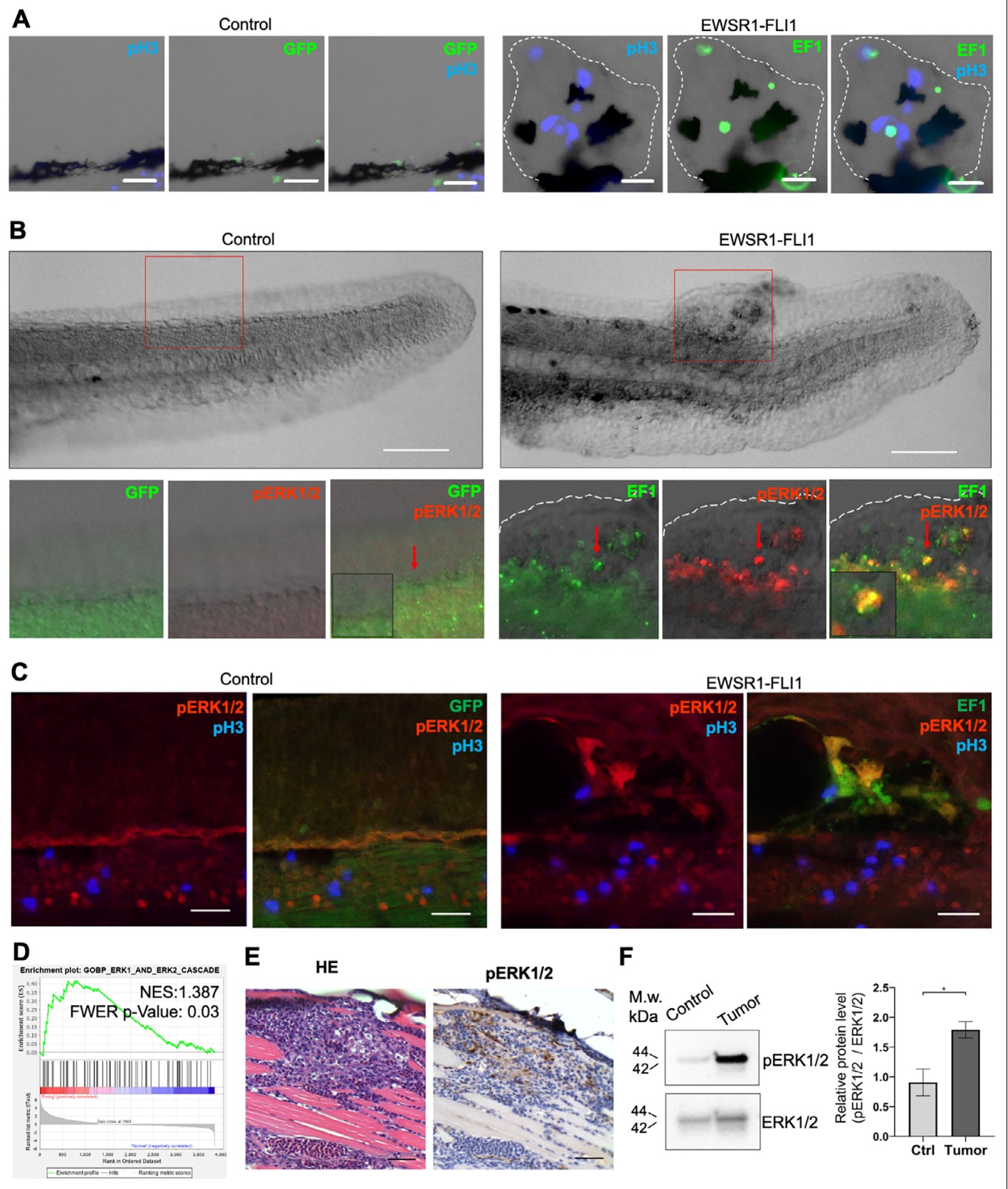

**Figure 4.** *EWSR1-FLI* expression activates the ERK1/2 signaling pathway in vivo. (**A**) Immunofluorescent staining of e*GFP* (control) and e*GFP2A-EWSR1-FLI1*-expressing embryos at 48 hpf. Blue: phosphohistone H3. Green: eGFP or eGFP2A-EWSR1-FLI1. A small region of the dorsal surface is shown. The outgrowth is outlined with a dashed white line. Scale bars, 20 μm. (**B**) Immunofluorescent staining of e*GFP* (control) and e*GFP2A-EWSR1-FLI1*-expressing embryos at 24 hpf for pERK1/2 (red) and eGFP. Scale bars, 100 μm. (**C**) Immunofluorescent staining of e*GFP* (control) and e*GFP2A-EWSR1-FLI1*-

*Figure 4 continued on next page*

Figure 4 continued

expressing embryos at 48 hpf for pERK1/2 (red), pH3 (blue), or eGFP (green). Scale bars, 20 μm. (**D**) Gene set enrichment analysis (GSEA) showing the enrichment of genes associated with ERK1/2 pathway (GOBP_ERK1_AND_ERK2_CASCADE) in zebrafish tumors. (**E**) Immunostaining of zebrafish tumor for pERK1/2. (**F**) Immunoblot analysis and immunoblot quantification of pERK1/2 and ERK1/2 levels in tumor and normal tissue. Error bars represent standard error of the mean (SEM), $N = 3$, *p < 0.05, two-tailed Student's *t*-test.

The online version of this article includes the following figure supplement(s) for figure 4:

**Figure supplement 1.** Immunostaining of zebrafish tumors for pERK1/2.

*Figure 4—figure supplement 1B*), complemented by immunoblot analysis (*Figure 4F*) (*N* of repeats = 3, *N* of replicas = 3). Similar to *EWSR1-FLI1*-driven outgrowth in larvae, ERK1/2 is active in advanced zebrafish tumors. ERK1/2 signaling activity in tumors was heterogeneous, with focal areas showing *Video 1* more intense signaling activity.

## EWSR1-FLI1 affects proteoglycan metabolism in zebrafish embryonic model

To gain further insight into the impact of *EWSR1-FLI1* expression during early tissue and organ development and the mechanisms driving increased growth factor signaling and cell proliferation, we performed MS analysis. As above, we integrated the *ubi:RSG-EWSR1-FLI1* cassette into the zebrafish genome in the presence of *Cre* mRNA. The control group of embryos was coinjected with an *ubi:RSG* transposon and Cre mRNA. Embryos were sorted for eGFP expression at 24 and 48 hpf time points and used to generate samples for LC–MS/MS analysis. Each sample consisted of 10 embryos (*N* of replicas = 3). EWSR1-FLI1 significantly affects protein expression in zebrafish embryos (*Figure 5A*). We identified 248 differentially expressed proteins affected by EWSR1-FLI1 at 24 hpf, and 1102 at 48 hpf (*Figure 5B*). Downregulated proteins comprised 65% and 73% of differentially expressed proteins at 24 and 48 hpf, respectively, suggesting that EWSR1-FLI1 was acting mostly as a transcriptional repressor rather than an activator (*Figure 5B*).

Gene ontology (GO) enrichment analysis of proteomics data at the 48 hpf time point showed that downregulated proteins were involved in regulation of cell metabolism (*Figure 5C*). Importantly, we found a strong upregulation of proteins involved in extracellular matrix (ECM) reorganization, proteoglycan metabolism, and protein synthesis (*Figure 5D*). The proteomics data revealed that *EWSR1-FLI1* expression in zebrafish embryos is associated with upregulated expression of collagens *col1a1b*, *col1a2*, *col2a1a*, *col9a1b*, *col9a2*, and *col28a2a*, which are involved in ECM organization and skeletal system development. Proteins involved

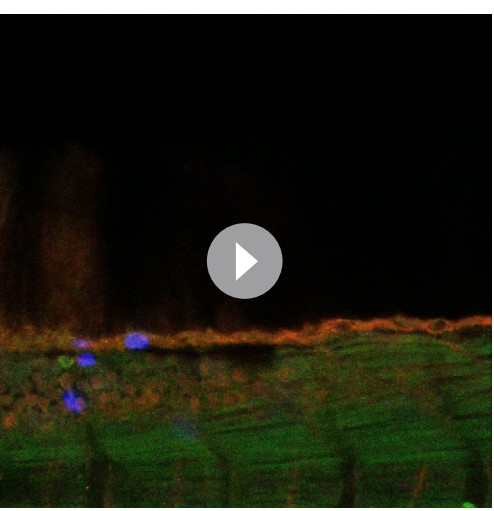

**Video 1.** Immunofluorescence staining of a 48 hpf zebrafish embryo for pERK1/2 (red), EWSR1-FLI1 (green), and pH3 (blue). Confocal Z-stack focusing on a region of the trunk and dorsal fin.

https://elifesciences.org/articles/69734/figures#video1

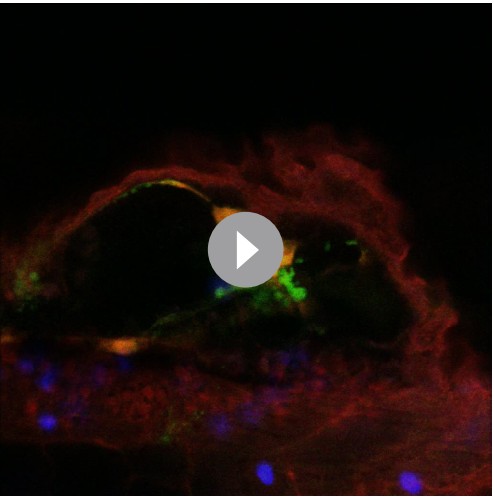

**Video 2.** Immunofluorescence staining of a tumor outgrowth in 48 hpf zebrafish embryo for pERK1/2 (red), EWSR1-FLI1 (green), and pH3 (blue). Confocal Z-stack focusing on an outgrowth arising from the dorsal surface of the embryo.

https://elifesciences.org/articles/69734/figures#video2

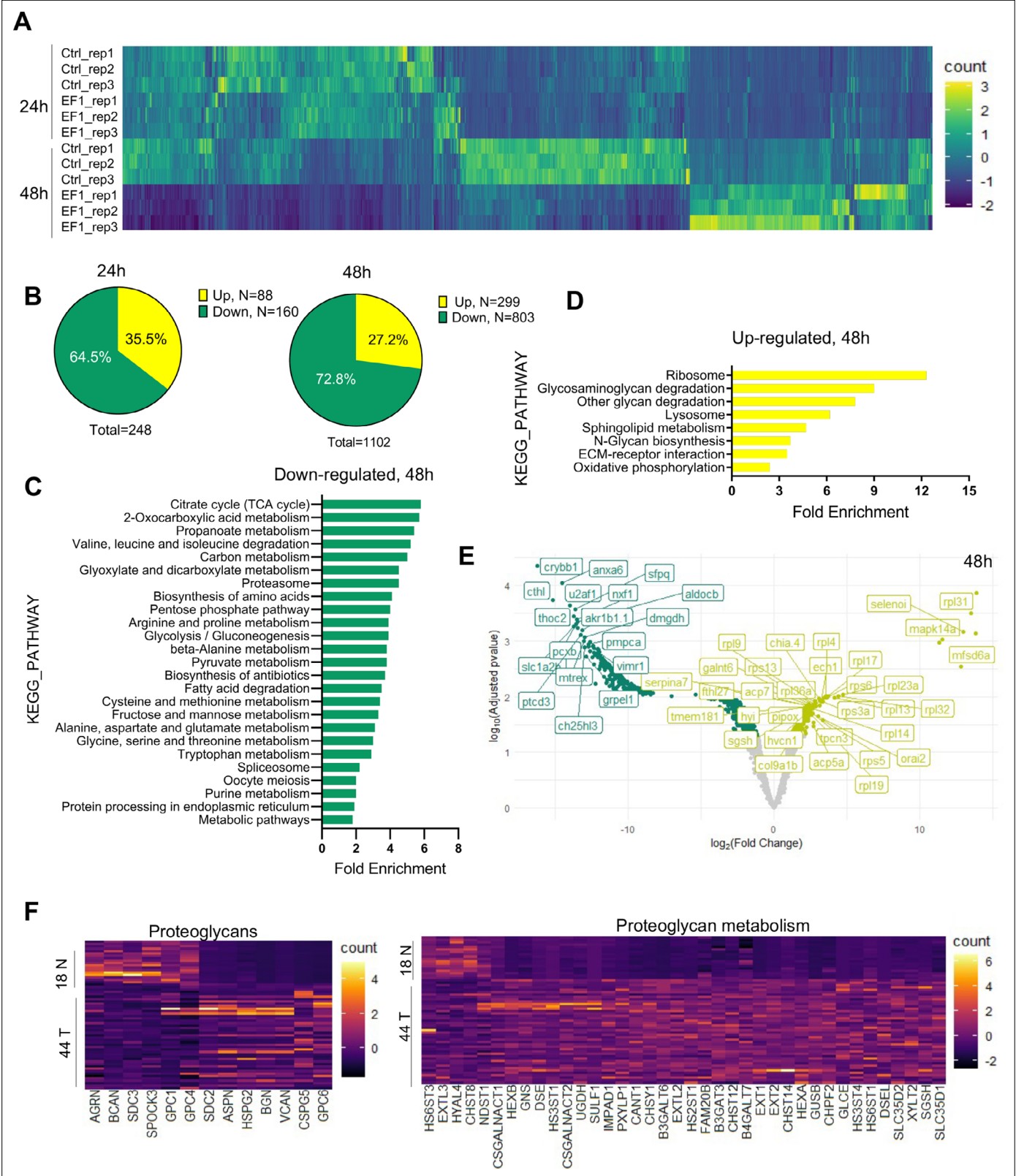

**Figure 5.** Liquid chromatography–mass spectrometry (LC–MS/MS) analysis of proteins affected by *EWSR1-FLI1* expression in developing zebrafish. (**A**) Heat map representing the differentially expressed proteins dysregulated by *EWSR1-FLI1* at 24 and 48 hpf. (**B**) Quantitative analysis of the differentially expressed proteins at 24 and 48 hpf. (**C, D**) Gene ontology (GO) analysis of differentially expressed proteins at 48 hpf. (**E**) Volcano plot of significantly downregulated and upregulated proteins in *EWSR1-FLI1*-expressing embryos. (**F**) Heatmap of the most differentially expressed proteins involved in proteoglycan metabolism in human Ewing sarcoma compared to normal tissue (GSE17674).

in proteoglycan catabolism were also significantly differentially expressed in embryos expressing *EWSR1-FLI1* (*Figure 5E*). Among the top identified hits was *N*-sulfoglucosamine sulfohydrolase (*sgsh*), the enzyme involved in heparan sulfate proteoglycan catabolism (*Figure 5E*). We also found upregulated *gnsb* and *gnsa* (*N*-acetylglucosamine-6-sulfatase) enzymes involved in hydrolysis of heparan sulfate chains.

To compare results obtained from proteomics data of Ewing sarcoma zebrafish model with those from human Ewing sarcoma, we analyzed the profile of proteins associated with proteoglycan metabolism from microarray data of 44 Ewing sarcoma samples and 18 normal tissue samples (GSE17674). Expression of enzymes involved in proteoglycan metabolism was strongly upregulated in tumor samples compared to normal tissue (*Figure 5F*). Specifically, we found upregulation of enzymes involved in the catabolism of heparan sulfate proteoglycans (including *SGSH*, *GNS*, *HS3ST4*, *HS3ST1*, *HS2ST1*, and *HS6ST1*) in human Ewing sarcoma (*Figure 5F*). Furthermore, proteoglycan metabolism was also strongly dysregulated in Ewing sarcoma tissues.

Taken together, our genetic model of Ewing sarcoma revealed that EWSR1-FLI1 expression dysregulates normal protein expression starting from early zebrafish development. The expression of EWSR1-FLI1 is associated with strong upregulation of ECM proteins and, most notably, enzymes involved in heparan sulfate proteoglycan catabolism. Supporting these data, we show that the genes involved in proteoglycan metabolism are strongly upregulated in human Ewing sarcoma tumors.

## Surfen inhibits proteoglycan-mediated activation of ERK1/2

Proteoglycans play a key regulatory role in the interactions of cells with ECM proteins, growth factors, and cytokines, and thus have major influence on cell signaling pathways that directly affect cancer growth (*Edwards, 2012*; *Iozzo and Sanderson, 2011*; *Multhaupt et al., 2016*; *Mythreye and Blobe, 2009*). Our finding that *EWSR1-FLI1* expression is associated with altered proteoglycan expression and metabolism suggests that dysregulation of proteoglycans may contribute significantly to growth promoting signaling pathways in Ewing sarcoma, including ERK signaling. Thus, proteoglycan metabolism could serve as a novel target for Ewing sarcoma. To test these possibilities, we performed treatment of Ewing sarcoma cell lines with the small molecule surfen (*bis*-2-methyl-4-amino-quinolyl-6-carbamide). Surfen is a sulfated heparan sulfate antagonist with a high affinity to all sulfated GAGs (*Schuksz et al., 2008*). Surfen blocks sulfation and degradation of GAG chains in vitro and affects growth factor binding and proteoglycan-mediated signal transduction through certain cell surface growth factor receptors (*Schuksz et al., 2008*).

First, we tested whether surfen can block proteoglycan-mediated activation of ERK1/2 signaling in the TC32 and EWS502 Ewing sarcoma cell lines. To reduce the basal level of ERK1/2 phosphorylation we performed pretreatment of cells with serum-free media for 4 hr, and then replaced serum-free media with the full-media in the presence of 1% DMSO or 2.5, 5, or 10 µM surfen. After 30 min of treatment, cell lysates were prepared and analyzed by immunoblotting. Samples were normalized to the value of total ERK1/2. As expected, the introduction of serum led to the activation of ERK1/2 signaling (*Figure 6, B*). Addition of 5 or 10 µM surfen strongly significantly inhibited ERK1/2 phosphorylation (*Figure 6A, B*). Thus, blocking proteoglycan metabolism impairs ERK1/2 signaling in Ewing sarcoma cells.

We next tested the effect of surfen treatment on the proliferation rates of TC32 and EWS502 cells exposed to surfen or DMSO vehicle. Treatment with surfen significantly reduces the proliferation rates of Ewing sarcoma cells (*Figure 6C*). Importantly, treatment of cells with surfen at 10 µM resulted in a decrease of TC32 proliferation by 62.7%, and EWS502 by 63.4% compared to control cells treated with 0.2% DMSO (*Figure 6C*). Surfen treatment caused morphological changes in TC32 and EWS502 cells (*Figure 6D*), consistent with predicted effects on cell adhesion.

To estimate cell survival and the ability of a single cell to grow into a colony we performed a clonogenic assay under low-adhesive conditions. We seeded 500 cells/well in a 24-well low-adhesive plate treated with surfen or 1% DMSO, under serum-deprived conditions. Serum was added 12 hr later to stimulate growth factor-mediated signaling. The colony number was analyzed after 2 or 3 weeks of incubation for TC32 and EWS502, respectively. Treatment of cells with surfen led to a significant reduction in colony number for both cell lines (*Figure 6, F*). More specifically, the treatment of TC32 cell with 2.5 µM surfen led to a 90.3% decrease of colony number while treatment of EWS502 cells with 2.5 µM surfen resulted in a 97.8% decrease of colony forming units. Thus, surfen targets the

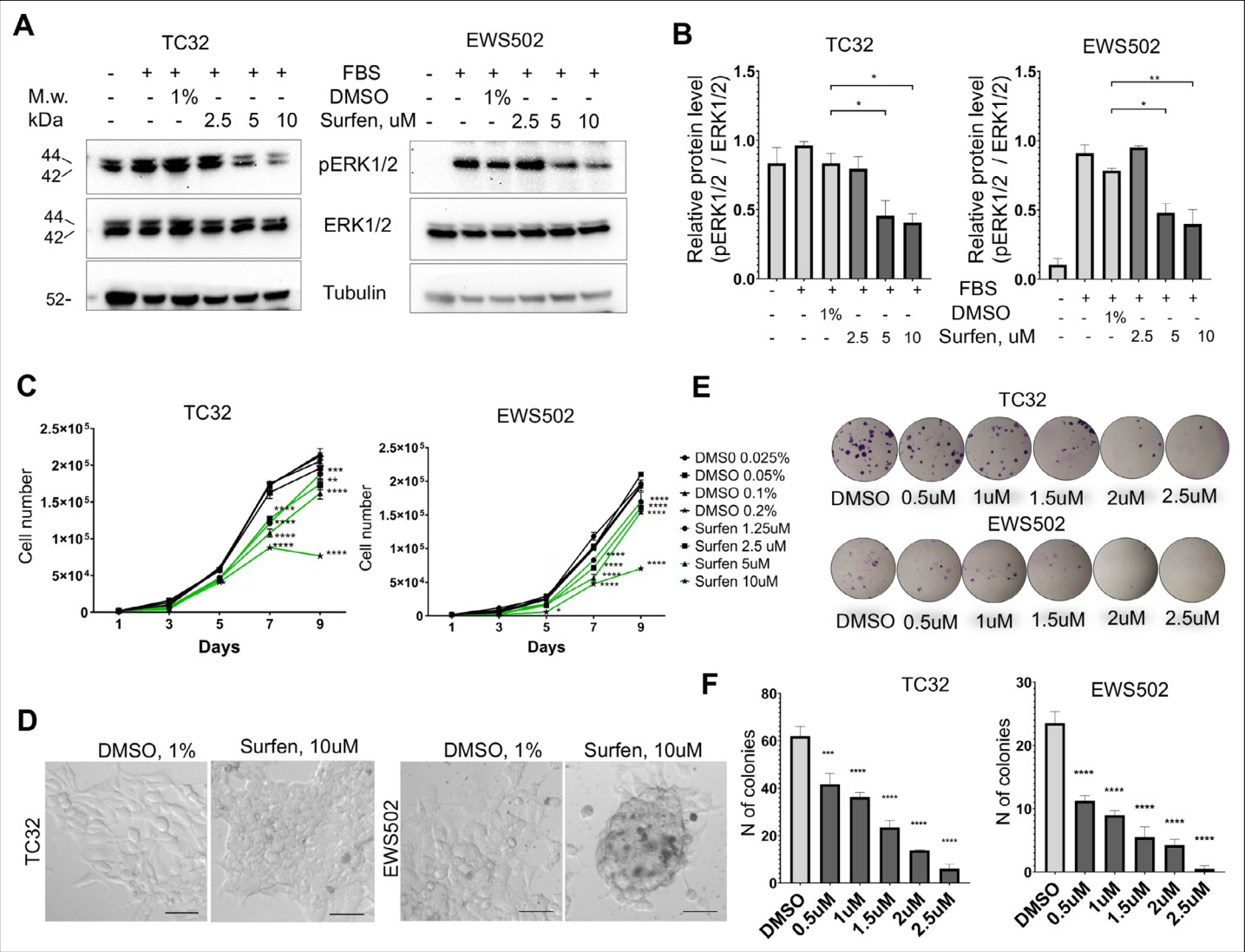

**Figure 6.** Surfen treatment impairs ERK1/2 signaling and growth of Ewing sarcoma cells. (**A**) Immunoblot analysis of pERK1/2, ERK1/2, and tubulin levels in TC32 and EWS502 cells treated with surfen or DMSO. (**B**) Immunoblot quantification of pERK1/2 expression level relative to ERK1/2 in TC32 and EWS502 cells treated with surfen or DMSO vehicle control. Error bars represent standard error of the mean (SEM), N = 3, *p < 0.05, **p < 0.01, based on one-way analysis of variance (ANOVA) test. (**C**) Proliferation rates of TC32 and EWS502 cells treated with surfen or DMSO. Error bars represent SEM, N = 3, **p < 0.01, ***p < 0.001, ****p < 0.0001 based on one-way ANOVA test. (**D**) Morphological changes in TC32 and EWS502 cells after surfen or DMSO treatment. (**E**) Clonogenic assay and (**F**) clonogenic assay quantification of TC32 and EWS502 cells treated with surfen or DMSO. Error bars represent SEM, N = 4, ***p < 0.001, ****p < 0.0001 based on one-way ANOVA test.

heparan sulfate proteoglycan-mediated activation of ERK1/2 signaling in Ewing sarcoma cells, inhibiting cancer cell proliferation and cell survival.

## Surfen inhibits EWSR1-FLI1-mediated growth in zebrafish model

We next tested whether surfen could inhibit the formation of *EWSR1-FLI*-driven outgrowths in vivo in our zebrafish model. As described above, we generated fish mosaically expressing *EWSR1-FLI1*. Fish with eGFP-positive outgrowths were identified at 24 hpf and treated with 0.2 μM surfen or 0.2% DMSO. Fish were imaged after 24 and 48 hr of treatment (*Figure 7A*). Consistent with results on human cells, surfen inhibited outgrowth development in the zebrafish embryo model of Ewing sarcoma.

Next, we determined whether surfen and MEK inhibitor trametinib have comparable effects on outgrowth formation (*Figure 7—figure supplement 1*). To determine the experimental concentration of trametinib, we treated embryos at 24 hpf with 1, 10, or 20 μM trametinib, as well as 0.2 μM,

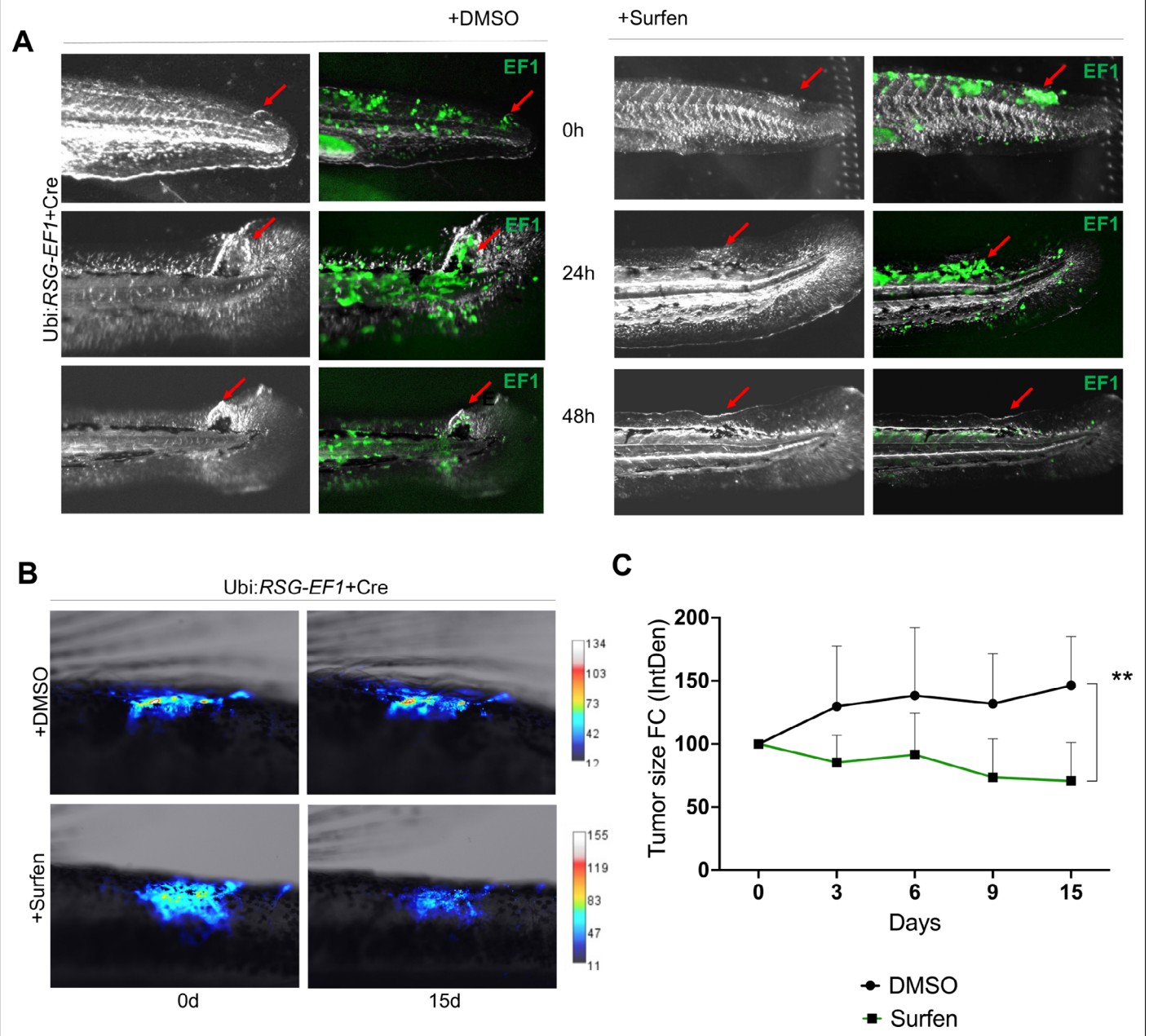

**Figure 7.** Surfen inhibits *EWSR1-FLI1*-mediated growth in the zebrafish model. (**A**) Surfen inhibits the development of *EWSR1-FLI1*-driven outgrowths in zebrafish larvae. (**B**) Surfen inhibits progression of *EWSR1-FLI1*-driven tumors. Animals with eGFP-positive tumors were grouped for treatment with surfen (*N* = 4) or DMSO (*N* = 4). Each group had tumors similar in size and location. (**C**) Quantification of changes in tumor size based on fluorescence intensity. Error bars represent SD, **p < 0.01 based on two-way analysis of variance (ANOVA) test.

The online version of this article includes the following figure supplement(s) for figure 7:

**Figure supplement 1.** Targeting ERK signaling pathway in embryonic model of Ewing sarcoma.

**Figure supplement 2.** Changes in the fin shape driven by *EWSR1-FLI1* expression as a readout for drug screening.

**Figure supplement 3.** Surfen inhibits progression of EWSR1-FLI1-driven tumors.

1 μM surfen, or 0.25% DMSO for 96 hr. We found that survival rate of embryos treated with 1 and 10 μM trametinib was not significantly different from survival rate of embryos treated with 0.2 μM surfen or 0.05% DMSO (*Figure 7—figure supplement 1A*). Next, we tested whether treatment of embryos with 1 μM trametinib and 0.2 μM surfen have effects on ERK phosphorylation. We treated WT embryos at 24 hpf with 1 μM trametinib, 0.2 μM surfen or 0.05% DMSO for 24 hr. Embryos were

used for western blot analysis. We found that trametinib strongly decreased the total level of ERK1/2 phosphorylation (*Figure 7—figure supplement 1B*). In contrast, surfen did not affect the total level of ERK1/2 phosphorylation in normal fish (*Figure 7—figure supplement 1B*), likely reflecting the fact that not all MAPK signaling is controlled by proteoglycans. To look specifically at cells expressing *EWSR1-FLI1* we generated fish mosaically expressing *EWSR1-FLI1*, sorted embryos at 24 hpf based on eGFP expression, and treated them with 0.2 μM surfen, 1 μM trametinib, or 0.05% DMSO. After 24 hr of treatment we fixed embryos in 4% PFA and performed immunofluorescent staining for eGFP and pERK1/2. Consistent with our previous data, we found that cells expressing EWSR1-FLI1 are positive for pERK1/2, however treatment of embryos with 0.2 μM surfen or 1 μM trametinib abrogates ERK1/2 phosphorylation in EWSR1-FLI1-positive cells (*Figure 7—figure supplement 1C*).

To compare the effect of MEK inhibitor trametinib and surfen on outgrowth formation we generated fish mosaically expressing *EWSR1-FLI1* and sorted them based on the presence of eGFP-positive outgrowths at 24 hpf. Prior the treatment we documented the eGFP fluorescence in all outgrowths. We treated fish with outgrowths with 0.2 μM surfen, 1 μM trametinib, or 0.05% DMSO and analyzed the size of the outgrowth 24 hr later. We found that treatment of outgrowths with 1 μM trametinib or 0.2 μM surfen has comparable effects on outgrowth formation (*Figure 7—figure supplement 1D*).

To quantify the effect of surfen, we took advantage of the changes in fin morphology driven by the EWSR1-FLI oncofusion during the first 3 days of zebrafish development. We noticed that *EWSR1-FLI1* expression is reproducibly associated with distortion of the normally straight appearance of the dorsal fin border (*Figure 7—figure supplement 2A*). To quantify this effect we calculated the coefficient of curvature (Ccurv), which measures the degree of deviation of the border from a straight line; more irregular fins are characterized by higher Ccurv (*Figure 7—figure supplement 2B, C*). We hypothesized that treatment of fish with drugs targeting *EWSR1-FLI1*-related pathways would result in the rescue of the fin phenotype (*Figure 7—figure supplement 2B*).

Applying this metric, surfen did not affect the fin shape of control fish (*Figure 7—figure supplement 2D*). *EWSR1-FLI1* expression strongly increased the Ccurv, and surfen treatment of fish expressing *EWSR1-FLI1* resulted in significant rescue of the phenotype (*Figure 7—figure supplement 2D*).

Finally, we aimed to investigate how surfen affects established tumors in our genetic model of Ewing sarcoma. We generated fish mosaically expressing *EWSR1-FLI1* and sorted animals with *EWSR1-FLI1*-induced, eGFP-positive tumors 2 months later. We split tumor-bearing animals into two balanced groups for treatment with surfen (*N* = 4) or DMSO (*N* = 4) (see Methods for treatment scheme). Each group had tumors similar in size and location. Prior the treatment we documented the eGFP expression in all tumors. We found that surfen significantly reduced the tumor size by 30% during the first 15 days while tumors treated with DMSO showed increase in tumor size by 46% (*Figure 7B, C*; *Figure 7—figure supplement 3A, B*). Based on that we conclude that surfen is effective both preventing the outgrowth formation in embryonic model of Ewing sarcoma and tumor progression in older fish.

Taken together, these results suggest that surfen may affect Ewing sarcoma growth via modulation of ERK1/2 signaling. However, it is possible that the growth-suppressive effects of surfen are due to surfen's effects on other signaling pathways affected by heparan sulfate proteoglycans.

## Discussion

Although Ewing sarcoma was first described over 100 years ago, there are few if any molecularly targeted therapies for patients with metastatic or relapsed disease. Fewer than 30% of patients presenting with metastases survive for 5 years (*Riggi et al., 2021*). While great progress has been made using cells, xenografts and PDX models, the development of complementary Ewing sarcoma animal models remains crucial for the understanding of disease biology in the complex developmental microenvironment. The importance of tumor microenvironment and communication between cancerous and host cells has become evident as essential for tumor formation and invasion. A better understanding of such mechanisms will support the development of new approaches, targeting not just cancer cells, but the environment around them.

Previous attempts by multiple groups to generate a mouse model of Ewing sarcoma were complicated by the high embryonic toxicity of the driver oncofusion (*Minas et al., 2017*). Consistent with this experience, we have tested a panel of ubiquitous and tissue-specific promoters to drive *EWSR1-FLI1* expression in zebrafish embryos (*Figure 1—figure supplement 1*). We found that in most cases,

expression of the oncogene causes cellular apoptosis, embryonic lethality, or developmental defects. Cre-inducible expression of *EWSR1-FLI1* under *cmv* and *b-actin* ubiquitous promoters did not result in a high incidence of tumor development. This result supports the importance of the level and timing of *EWSR1-FLI1* expression for efficient modelling of tumorigenesis. Despite the fact that cmv, b-actin, and ubi promoters drive ubiquitous gene expression they have slightly different efficiency, tissue tropism and timing of expression. However, we discovered that Cre-inducible expression of human *EWSR1-FLI1* under the ubiquitin promoter was associated with less oncofusion toxicity in zebrafish larva. The pattern of Cre-induced *eGFP-2A-EWSR1-FLI1* expression was consistently different from Cre-induced eGFP expression driven by the same ubiquitin promoter (*Figure 3B*), suggesting that *EWSR1-FLI1* can be tolerated by certain cell types while being toxic for others.

Human Ewing sarcoma can occur in any part of the body in bone or soft tissue. It most commonly involves the pelvis and proximal long bones (*Riggi et al., 2021*). In our model, based on mosaic integration of human *EWSR1-FLI1* into the zebrafish genome, formation of tumors was observed in association with fish skeleton at the regions proximal to the base of pectoral, anal, and caudal fins, and at supraneurals. Fifty-eight percent of fish had more than one tumor (*Figure 1—figure supplement 3A*). Interestingly, we observed the formation of ectopic fins driven by *EWSR1-FLI1* (*Figure 1—figure supplement 3B*) suggesting that *EWSR1-FLI1* may potentially redirect the differentiation program of transformed cells. This model presents several advances over our previously reported zebrafish mosaic model of Ewing sarcoma (*Leacock et al., 2012*). In that model, on a wild-type background only 0.6% of fish developed tumors during the 15 months, and tp53 deficiency was required for more penetrant tumor formation (*Leacock et al., 2012*). We found that 40% of affected fish developed leukemia-like tumors rather than sarcomas. In our new Cre-inducible mosaic model, 34% of fish developed tumors and only 1 fish out of 77 developed a leukemia-like SRBCT (*Figure 1C*). These findings highlight that the level and spatiotemporal distribution of *EWSR1-FLI1* expression plays an important role in efficient tumor generation.

To characterize tumors, we performed IHC staining for markers commonly used in diagnosis of human Ewing sarcoma. Staining of zebrafish tumors with antibodies against FLI1 showed that cells have different expression level of *EWSR1-FLI1* (*Figure 2D*), consistent with recent reports of cell-to-cell heterogeneity which affects proliferative and migratory potential of Ewing sarcoma cells (*Franzetti et al., 2017*). While silencing of transgene expression in individual cells may occur, the fact that e*GFP* and *EWSR1-FLI1* are expressed from a single mRNA transcript, and e*GFP* expression is widely retained throughout the tumor, makes this possibility less likely. Staining for CD99 and PAS are widely used for Ewing sarcoma diagnosis in patients (*Muhammad et al., 2012*). CD99 is a cell surface transmembrane protein highly expressed in all Ewing's sarcomas. Zebrafish tumor cells were positive for CD99 (*Figure 2A*). Additionally, consistent with human data zebrafish tumors were positive for PAS which stains glycogen and polysaccharides enriched in Ewing sarcoma (*Figure 2B*). We showed that the expression of *nr0b1* in tumor tissue is strongly upregulated resembling the human phenotype (*Kinsey et al., 2006*). Altogether, zebrafish tumors are positive for known Ewing sarcoma markers recapitulating the main features of the human disease.

Previous studies established that 91% of human tumors had proliferating populations of cells positive for the proliferation marker, KI67. A high proliferation index was predictive of poor overall survival independent of tumor site, tumor volume, or metastasis at diagnosis (*Brownhill et al., 2014*). We identified increased proliferation in *EWSR1-FLI1*-driven outgrowths in the earliest stages of tumor formation. In a wide variety of cancers, the ERK1/2 signaling pathway is known to control cell proliferation. Moreover, it was shown that ERK1 and ERK2 are constitutively activated in NIH-3T3 cells expressing *EWSR1-FLI1* as well as in several human Ewing's sarcoma tumor-derived cell lines (*Silvany et al., 2000*). Consistent with these data, we identified activated phosphorylation of ERK1/2 in both *EWSR1-FLI1*-expressing outgrowths and tumors in the in vivo zebrafish model (*Figure 4*). In summary, tumorigenesis in fish is associated with activation of ERK1/2 signaling.

To study the mechanisms underlying the activation of ERK1/2 triggered by *EWSR1-FLI1* we performed LC–MS/MS analysis. We discovered that that *EWSR1-FLI1* affects expression of proteins involved in ECM reorganization and proteoglycan catabolism (*Figure 5A, C and D*). It is known that tumors leverage ECM remodeling to create a microenvironment that promotes tumorigenesis and metastasis. These tumor-driven changes support tumor growth, migration, and invasion. Our data show that expression of *EWSR1-FLI1* is associated with the strong production of collagens col1a1b,

col1a2, col2a1a, col9a1b, col9a2, and col28a2a both in embryos and in advanced tumors (data not shown), suggesting the importance of specific matrix for tumor development. We found a strong upregulation of proteins involved in proteoglycan catabolism in human Ewing sarcoma (*Figure 5F*). Interestingly, the upregulation of heparanase, the enzyme involved in cleavage of the side chains of heparan sulfate proteoglycans, was reported in Ewing sarcoma tumors correlating with poor prognosis in patients (*Shafat et al., 2011*). Thus, Ewing sarcoma oncogenesis is associated with dysregulation of proteoglycan turnover.

Proteoglycans are important components of extracellular matrix regulating Wnt, Hedgehog, TGF-β, FGFR, and other signaling pathways. Proteoglycans stabilize ligand–receptor interactions, creating potentiated ternary signaling complexes that regulate the signaling pathways involved in cell proliferation, migration, adhesion, and growth factor sensitivity (*Elfenbein and Simons, 2010*). For example, binding of FGF to its signaling receptor requires prior binding to the heparan sulfate side chains of the proteoglycans (*Mythreye and Blobe, 2009*). Thus, dysregulation of proteoglycan catabolism can result in aberrations in signal transduction from cell surface receptors.

To block the aberrantly activated proteoglycan-mediated pathways in Ewing sarcoma we targeted the binding of GAG chains with the ligands and signal molecules. Surfen (*bis*-2-methyl-4-aminoquinolyl-6-carbamide) binds to heparan sulfate and other GAGs blocking the sulfation and degradation of GAG chains in vitro (*Schuksz et al., 2008*). Surfen also affects responses dependent on heparan sulfate such as growth factor binding and activation of downstream signaling pathways (*Schuksz et al., 2008*). To target proteoglycan-mediated activation of ERK1/2 signaling we treated TC32 and EWS502 cells with surfen. Surfen impaired Ewing sarcoma cell proliferation at all doses tested, with the strongest effects at 5 and 10 µM (*Figure 6C*). Concomitant with the effect on proliferation, ERK1/2 phosphorylation was downregulated at these doses of surfen, indicating that Ewing sarcoma cell lines are very sensitive to surfen-mediated blockage of cell surface receptor signaling.

To test whether surfen is effective in vivo in the zebrafish model we treated fish with outgrowths and established tumors via aqueous exposure to surfen or DMSO. Surfen demonstrated remarkable inhibition of outgrowth development and tumor progression in genetic zebrafish model of Ewing sarcoma (*Figure 7A, B*). Thus, surfen is effective against human Ewing sarcoma cells in vitro and against tumor growth in the zebrafish model in vivo. Surfen was originally developed for use in humans to facilitate depot insulin deposition (*Umber et al., 1938*), and has been tested as an agent against glioblastoma tumor cells (*Logun et al., 2019*). Thus, targeting glycosaminoglycan metabolism may represent a new therapeutic opportunity for Ewing sarcoma. We quantified the effect of surfen in vivo using the fin coefficient of curvature Ccurv (*Figure 7—figure supplement 2*). As a complement to studies using tumor cell outgrowths, this flexible and quantifiable assay has great potential for high-throughput screening to identify additional drugs targeting proteoglycan-mediated signaling in Ewing sarcoma.

Overall, here we present a new inducible zebrafish model of Ewing sarcoma. The advantage of the model is the opportunity to study tumorigenesis in vivo, in a complex developmental background. The system further allows study of tumor cell behavior using high-resolution imaging and is effective for high-throughput drug screening. Our findings suggest that the temporal expression of *EWSR1-FLI1* is crucial for tumor development, supporting the existence of a specific cell or lineage of origin for Ewing sarcoma. Our new *EWSR1-FLI1* zebrafish model of Ewing sarcoma emphasizes the role of proteoglycans mediating ERK1/2 signaling and growth of tumor cells. Further investigation of the interactions between Ewing sarcoma and the tumor microenvironment in vivo can provide critical insights that may lead to new therapies of the disease.

## Materials and methods
### Zebrafish husbandry
*Danio rerio* were maintained according to industry standards in an AALAAC-accredited facility. WIK wild-type fish were obtained from the ZIRC Zebrafish International Resource Center (https://zebrafish.org).

### Plasmids and cloning
Human *EWSR1-FLI1* coding sequence was provided by Chris Denny, University of California-Los Angeles, USA (*Leacock et al., 2012*). The Gateway expression system (Invitrogen) was used for

generation of all constructs for expression in zebrafish (*Kwan et al., 2007*). *EWSR1-FLI1* flanked by attB2r site (at 5′ primer) and attB3 site (at 3′ primer) was cloned into a 3′ entry vector according to the provided protocol (*Kendall and Amatruda, 2016*). The plasmids containing a stop-dsRed-stop sequence were a generous gift from Eric Olson. Likewise dsRed-STOP-eGFP-2A-*EWSR1-FLI1* coding sequence flanked by attB1 and attB2 sites was cloned into a middle entry vector (*Kwan et al., 2007*). The *ubi* promoter was a kind gift from Len Zon (Addgene #27320) (*Mosimann et al., 2011*). The *fli1* promoter was provided by Nathan Lawson (Addgene #31160 and #26031), and the *mitfa* promoter by James Lister (Addgene #81234) (*Kendall et al., 2018*). The *beta actin* promoter, *cmv* promoter were used for plasmids generation and expression in zebrafish. The plasmids containing a eGFP-2A sequence were a kind gift from Steven Leach, and were subcloned into a middle entry Gateway expression system (*Kendall et al., 2018*). The destination vector pDestTol2pA2 and 3′ SV40 late poly A signal construct, were used for construct generation by an LR reaction with LR Clonase II Plus (Invitrogen) (*Kwan et al., 2007*). Transposase, Cre, and GFP RNAs were synthesized from plasmids pCS2FA, pCS2-Cre.zf, and pCS2-GFP accordingly using the mMessage mMachine kit (Applied Biosystems/Ambion, Foster City, CA). The constructs generated include: *beta-actin-eGFP2A-pA* (Genevieve Kendall), *beta-actin-eGFP2A-EWSR1-FLI1*, *ubi-eGFP-2A*, *ubi-eGFP2A-EWSR1-FLI1*, *cmv-eGFP-2A*, *cmv-eGFP2A-EWSR1-FLI1*, *fli-eGFP-2A*, *FLI-eGFP2A-EWSR1-FLI1*, *mitfa-eGFP-2A*, *mitfa-eGFP2A-EWSR1-FLI1*, *beta-actin-dsRed-stop-eGFP-2A-pA*, *beta-actin-dsRed-stop-eGFP-2A-EWSR1-FLI1*, *ubi-dsRed-stop-eGFP-2A*, *ubi-dsRed-stop-eGFP-2A-EWSR1-FLI1*, *cmv-dsRed-stop-eGFP-2A*, *cmv-dsRed-stop-eGFP-2A-EWSR1-FLI1*, *fli-dsRed-stop-eGFP-2A*, *fli-dsRed-stop-eGFP-2A-EWSR1-FLI1*, *mitfa-dsRed-stop-eGFP-2A*, and *mitfa-dsRed-stop-eGFP-2A-EWSR1-FLI1*.

## Zebrafish injections

Zebrafish embryos were injected at the single-cell stage. The injection mixes contained 50 ng/µl of Tol2 transposase mRNA, 10–50 ng/µl of described DNA constructs, 0.1% phenol red, and 0.3× Danieau's buffer. The injection mixes containing *beta-actin-dsRed-stop-eGFP-2A-pA*, *beta-actin-dsRed-stop-eGFP-2A-EWSR1-FLI1*, *ubi- dsRed-stop-eGFP-2A*, *ubi-dsRed-stop-eGFP-2A-EWSR1-FLI1*, *cmv-dsRed-stop-eGFP-2A*, *cmv- dsRed-stop-eGFP-2A-EWSR1-FLI1*, *fli-dsRed-stop-eGFP-2A*, *fli-dsRed-stop-eGFP-2A-EWSR1-FLI1*, *mitfa- dsRed-stop-eGFP-2A*, and *mitfa-dsRed-stop-eGFP-2A-EWSR1-*FLI1 also contained 0.5 ng of *Cre*_RNA or *GFP*_RNA.

## Zebrafish embryo survival

For survival analysis of embryos, fish were injected with *ubi:RSG-EWSR1-FLI1; Cre*_RNA or control mixes *ubi:RSG-EWSR1-FLI;GFP*_RNA *ubi:RSG;Cre*_RNA. The total number of injected fish was counted (the exact number of fish for each condition is indicated in the respective legend), and then the resulting alive embryos subsequently determined during first 10 days. Survival curves were plotted using GraphPad Prism 8.4.3. Biological replicates *N* = 3.

## Protein identification by LC–MS

Zebrafish embryos were injected with *ubi:RSG-EWSR1-FLI1;Cre*_RNA or *ubi:RSG;Cre*_RNA. Embryos were sorted for eGFP at 24 and 48 hpf time points. Sorted embryos were dechorionated, dissociated by pestle homogenizer in RIPA buffer supplemented with 1× inhibitor of proteases (Mini Protease Inhibitor Cocktail, cOmplete). Additionally, tumor or normal tissue from dorsal, caudal, and pectoral fins were dissected and processed in an analogous way. Protein levels were analyzed using the BCA protein assay Kit (Thermo Scientific). Lysates were heated at 95°C for 5 min and loaded on a 12% gel (BioRad). Gels were stained with Coomassie Blue. Stained 1 cm bands were cut out of gels, sliced into 1 mm cubes and transferred to 1.5 Eppendorf tubes for submission. Data were preproceeded and provided by the UT Southwestern Proteomics Core. Data were analyzed using an R package ROTS (*Suomi et al., 2017*) and visualized in R. GSEA was performed using the GSEA software according the instructions provided https://www.gsea-msigdb.org/gsea/index.jsp. The experiment was performed in three technical replicates for embryo samples and three biological replicates for tissue samples.

## Zebrafish tumor collection, processing for histology, and tumor incidence

Zebrafish were screened under a Nikon SMZ25 fluorescent stereomicroscope for the presence of eGFP-positive tumors. Fish with tumors were humanely euthanized and fixed in 4% PFA/1× phosphate-buffered saline (PBS) for 48 hr at 4°C. They were decalcified in 0.5 M EDTA (Ethylenediaminetetraacetic Acid) for 5 days, processed and mounted in paraffin blocks for sectioning and further experiments. For tumor incidence curves, zebrafish were injected and sorted for eGFP at 14 dpf and monitored for 4 months. Zebrafish with no eGFP fluorescence were considered as negative for *EWSR1-FLI*-dependent tumor formation. Zebrafish with tumors were collected and processed as described earlier for hematoxylin and eosin staining. All tumors were reviewed by an experienced sarcoma pathologist.

## Immunofluorescence

Embryos at 72 hpf were fixed overnight at 4°C in 4% PFA in 1× PBS and processed according to a published protocol for immunofluorescence staining (*Verduzco and Amatruda, 2011*). Zebrafish embryos were stained with primary antibodies directed against Phospho-p44/42 MAPK (Erk1/2) Thr202/Tyr204 (4370 S, Cell Signaling) at 1:200, GFP (4B10) (2955, Cell Signaling) at 1:300, or Phospho-Histone H3 (Ser10) (D7N8E) (53348, Cell Signaling) at 1:500. The secondary antibodies used were Goat anti-Mouse IgG (H + L) Alexa Fluor Plus 488 (# A32723, Thermo Fisher), Donkey anti-Rabbit IgG (H + L) Alexa Fluor 546 (A10040, Thermo Fisher), and Goat anti-Rabbit IgG(H + L) Alexa Fluor 405 (A31556, Thermo Fisher) at 1:500. The staining was repeated at least three times.

## IHC staining

Slides with paraffin embedded tissue sections were baked for 60 min at 60°C, immersed with xylene, 100% ethanol, 95% ethanol, 75% ethanol, distilled $H_2O$ two times each for 5 min each. Antigen retrieval was performed in Trilogy reagent (920 P, Sigma) for 10 min in the pressure cooker. Slides were cooled and blocked with 3% $H_2O_2$ for 30 min, followed by blocking with 1%BSA/1× PBST for 1 hr. Slides were incubated with primary antibodies Anti-CD99 antibody (ab108297, Abcam) at 1:200, Anti-FLI1 (ab15289, Abcam) at 1:100, anti-Phospho-p44/42 MAPK (Erk1/2) Thr202/Tyr204 (4370 S, Cell Signaling) at 1:200 overnight. Secondary antibodies used were Anti-rabbit IgG, HRP-linked Antibody (7074 S, Cell Signaling), Anti-mouse IgG, HRP-linked Antibody (7076 S, Cell Signaling). SignalStain DAB Substrate Kit #8059 was used for chromogen staining according to the manufacturer's instructions. Slides were also stained with hematoxylin and eosin, dehydrated, and mounted with permount mounting media. The staining was repeated more than three times.

## Imaging

Embryos were imaged at 12 hpf, 24 hpf, 48 hpf, 72 hpf, and 5 dpf on Nikon SMZ25 fluorescent stereomicroscope. Images of whole mount immune-stained zebrafish were taken on a Keyence BZ-X700 fluorescent microscope and Leica STELLARIS 5. Slides were imaged on a Keyence BZ-X700 fluorescent microscope, Leica DM4000B and Zeiss LSN710.

## RNA extraction, cDNA synthesis, and qRT-PCR

Fresh eGFP-positive tumor tissues or sorted dechorionated embryos were snap frozen in liquid nitrogen. Frozen tissues and embryos were subjected to total RNA isolation using the RNeasy Microkit manufacturer's instructions (Qiagen). The RT2 HT First Strand Synthesis kit (QIAGEN) was used for cDNA reverse transcription from 200 ng to 1 μg of total RNA. qRT-PCR (quantitative Real Time Polymerase Chain Reaction) was performed on a CFX384 Touch Real-Time PCR Detection System using the SYBRGreen Master Mix (BioRad). See for primer sequences. Error bars indicate standard deviation. To determine significance a Student's *t*-test was performed on normalized Ct replicates. The RT-PCR analysis was performed at least three times for each experiment in three technical replicates for each condition.

## Whole mount in situ hybridization

For the in situ hybridization, we used the RNA scope approach (*Wang et al., 2012*). Embryos at 24 hpf were fixed in 4% PFA and used for double RNAscope staining for nkx2.2a (529751-C2 RNAscope Probe – Dr-nkx2.2a-C2) and eGFP (538851 RNAscope Probe – EGFP-O4). Staining was performed

based on the RNAscope assay on Whole Zebrafish embryos protocol with the following modifications. All hybridization steps were performed in a 40°C water bath. Samples were washed twice each washing step using 1 ml of 1× Wash Buffer for 5 min each time with gentle shaking.

## Cell culture

TC32 and EWS502 cell lines were the kind gift of Dr. Angelique Whitehurst. All cell lines were authenticated by STR genotyping and regularly confirmed to be free of mycoplasma contamination. TC32 was maintained in RPMI 1640 GlutaMAX with 10% fetal bovine serum (Sigma) and 1× Antibiotic–Antimycotic (Gibco) at 37°C in 5% $CO_2$. EWS502 was maintained in RPMI 1640 GlutaMAX with 15% FBS (Sigma) and 1× Antibiotic–Antimycotic (Gibco) at 37°C in 5% $CO_2$.

## Surfen treatment

Surfen hydrate, ≥98% was obtained from Sigma-Aldrich (S6951). Because Surfen binds avidly to plastic, it is necessary to use glass vessels or precoat all plasticware with serum before use. Surfen stock solutions were prepared in DMSO (dimethyl sulfoxide) at 1, 4, and 5 mM, aliquoted and stored in glass containers at −20°C in the dark.

## Zebrafish treatment with surfen and trametinib

For zebrafish treatment working solutions were prepared by diluting stock solutions of surfen (1 and 4 mM) or trametinib (20 and 100 mM) in E3 medium. Final concentrations for treatments were determined as 0.2 µM for surfen hydrate (S6951, Sigma-Aldrich) and 1 µM for trametinib (GSK1120212, Selleck Chemicals) (*Figure 7—figure supplement 1A*). Embryos were injected, screened for eGFP at 24 hpf, dechorionated and incubated in 20 ml of E3 medium containing appropriate amount of DMSO or surfen/trametinib in a glass beaker. All solutions were changed daily. Biological replicates $N = 4$. The exact number of fish for each condition is indicated on figure legend.

The scheme of treatment for zebrafish with tumors included 6 days with 0.2 µM surfen, 3 days with 0.8 µM surfen, 3 days drug holiday with no drug, and 3 days with 0.8 µM surfen. Control group was treated with DMSO. Total duration of the experiment was 15 days.

## Protein extraction and western blots

Cells or tumor tissues were harvested and lysed in RIPA buffer complemented with cOmplete Mini Protease Inhibitor Cocktail inhibitors and Phosphatase inhibitor cocktails (Sigma). Lysates were denatured by boiling in 1× Laemmli buffer at 95°C for 5 min and loaded on a 4–20% gradient gel (BioRad). PVDF (polyvinylidene difluoride) membranes were used for proteins wet transfer (BioRad). Membranes were blocked in Casein + 0.1% Tween-20 (Thermo), and incubated overnight at 4°C with the primary antibodies Phospho-p44/42 MAPK (Erk1/2) (Thr202/Tyr204) (4370 S, Cell Signaling) at 1:2000, p44/42 MAPK (Erk1/2) (137F5) (4695 S, Cell signaling) at 1:2000, β-Actin Antibody (4967 S, Cell Signaling) at 1:2000, and α-Tubulin Antibody (2144 S, Cell Signaling) at 1:2000. Secondary antibody used were Anti-rabbit IgG, HRP-linked Antibody (7074 S, Cell Signaling) at 1:5000, and Anti-mouse IgG, HRP-linked Antibody (7076 S, Cell Signaling) at 1:5,000. Signal was detected using SuperSignal West Pico Chemiluminescent Substrate (Fisher). BioRad GelDoc XR was used for membranes imaging. The western blot analysis was performed at least three times for each experiment in three technical replicates.

## Cellular proliferation assays

For proliferation TC32 and EWS502 were seeded at 2000 cells per well of 96-well plate with three replicas per time point in 10% and 15% RPMI 1640 GlutaMAX media supplemented with 1× Antibiotic–Antimycotic (Gibco). Cells were treated with 1.75, 2.5, 5, and 10 µM of Surfen or 0.025%, 0.05%, 0.1%, and 0.2% DMSO on second, fourth, sixth, and eighth days. The number of cells at each point was measured using the hemocytometer at first, third, fifth, seventh, and ninth days. The proliferation assay was performed at least two times. Three technical replicates were performed per condition.

## Colonyformation assay

For colony formation assay 500 of TC32 and EWS502 cells were seeded per well of 24-well low-adhesive plate treated with 0.5, 1, 1.5, 2, and 2.5 µM of surfen or 1% DMSO under serum-deprived conditions. Serum was added 12 hr later. Medium with surfen or DMSO was changed three times a

week. The colony number was analyzed on days 14 and 21 for TC32 and EWS502 accordingly. For imaging media was removed, cells were prefixed and stained with 0.5% crustal violet. The colony formation assay was performed at least two times. Four technical replicates were performed per condition.

## Ccurvature calculation

To measure the distortion of the fin we introduced the coefficient of curvature $Ccurv = (L1 - L0)/L0 \times 100$, where $L1$ is the length of the fin border between points A and B and $L0$ is the length of the straight line between those two points (*Figure 7C*).

## Statistics

Statistical analysis for embryonic survival, tumor incidence curves, proliferation curves, colony formation, and relative gene expression, were performed using GraphPad Prism 8.4.3 (La Jolla, CA). The number of individual experiments, replicas, and samples analyzed, and significance is reported in the figure legends. Statistical significance was calculated by Student's *t*-test for two-group comparison, one-way analysis of variance for comparison of multiple groups with one control group and for comparison between different experimental groups. $p > 0.05$ = n.s., $*p < 0.05$, $**p < 0.01$, $***p < 0.001$, and $****p < 0.0001$.

## Acknowledgements

We thank the Children's Hospital Los Angeles Pathology and Cellular Imaging Cores, the UT Southwestern Medical Center Proteomics Core, and the University of Southern California High-Performance Computing Cluster for exceptional services and for their expertise. We grateful to Genevieve Kendall for productive discussions and help. JFA was supported by the Nearburg Professorship of Pediatric Oncology Research at UT Southwestern Medical Center during early stages of this work.

## Additional information

### Funding

| Funder | Grant reference number | Author |
| --- | --- | --- |
| National Cancer Institute | U54CA231649-01 | James F Amatruda |
| 1 Million 4 Anna | | James F Amatruda |

The funders had no role in study design, data collection, and interpretation, or the decision to submit the work for publication.

### Author contributions

Elena Vasileva, Conceptualization, Data curation, Formal analysis, Investigation, Methodology, Validation, Visualization, Writing - original draft, Writing – review and editing; Mikako Warren, Formal analysis, Investigation, Validation, Writing – review and editing; Timothy J Triche, Formal analysis, Validation, Writing – review and editing; James F Amatruda, Conceptualization, Data curation, Formal analysis, Funding acquisition, Investigation, Methodology, Project administration, Supervision, Validation, Writing – review and editing

### Author ORCIDs

Elena Vasileva http://orcid.org/0000-0003-2026-1331
James F Amatruda http://orcid.org/0000-0002-9901-2137

### Ethics

This study was performed in strict accordance with the recommendations in the Guide for the Care and Use of Laboratory Animals of the National Institutes of Health. All of the animals were handled according to approved institutional animal care and use committee (IACUC) protocols of the University of Southern California, protocol number 21,150.

Decision letter and Author response
Decision letter https://doi.org/10.7554/eLife.69734.sa1
Author response https://doi.org/10.7554/eLife.69734.sa2

## Additional files

### Supplementary files

• Supplementary file 1. Detailed characterization of zebrafish tumors driven by EWSR1-FLI1 expression.

• Supplementary file 2. List of downregulated and upregulated proteins identified by LC–MS/MS analysis in EWSR1-FLI1-expressing tumors, $p < 0.05$.

• Supplementary file 3. List of downregulated and upregulated proteins identified by LC–MS/MS analysis in EWSR1-FLI1-expressing embryos, $p < 0.05$.

• Supplementary file 4. List of primers for RT-PCR.

• Source data 1. Western Blot - Raw Data.

• Transparent reporting form

### Data availability

Proteomics data have been deposited in Dryad https://doi.org/10.5061/dryad.x95x69pj8.

The following dataset was generated:

| Author(s) | Year | Dataset title | Dataset URL | Database and Identifier |
|---|---|---|---|---|
| Vasileva E, Warren M, Triche T, Amatruda J | 2021 | Proteomics analysis of normal and EWSR1-FLI1-expressing embryos/tissue | https://doi.org/10.5061/dryad.x95x69pj8 | Dryad Digital Repository, 10.5061/dryad.x95x69pj8 |

The following previously published datasets were used:

| Author(s) | Year | Dataset title | Dataset URL | Database and Identifier |
|---|---|---|---|---|
| Savola S, Klami A, Myllykangas S, Manara C, Scotlandi K, Picci P, Knuutila S, Vakkila J | 2009 | Inflammatory gene profiling of Ewing sarcoma family of tumors (set B) | https://www.ncbi.nlm.nih.gov/geo/query/acc.cgi?acc=GSE17674 | NCBI Gene Expression Omnibus, GSE17674 |
| Khan J, Shern J, Langenau D | 2017 | Gene Expression in Human Rhabdomyosarcoma | https://www.ncbi.nlm.nih.gov/geo/query/acc.cgi?acc=GSE108022 | NCBI Gene Expression Omnibus, GSE108022 |

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
