## [Editor Report]

This model represents an improvement upon previous zebrafish sarcoma models and the data suggest that the methods employed yield tumors that resemble human disease. This new model may be used to better understand sarcoma progression so that new therapeutic targets may be realized.

---

## [Decision Letter]

**Decision letter after peer review:**

Thank you for submitting your article "Dysregulated heparan sulfate proteoglycan metabolism promotes Ewing sarcoma tumor growth" for consideration by *eLife*. Your article has been reviewed by 3 peer reviewers, one of whom is a member of our Board of Reviewing Editors, and the evaluation has been overseen by Richard White as the Senior Editor. The reviewers have opted to remain anonymous.

Essential revisions:

Three experts with have reviewed this manuscript and have made a number of suggestions which should be addressed. It was felt that the study may be considered for publication in *eLife*, pending the following essential revisions:

1) The comparison of the zebrafish tumours to human disease, should be enhanced. For example, protein levels in human samples should be compared to protein levels in the fish model, particularly since mRNA levels may not correspond to proteins. Alternatively, RNA levels in the fish could be compared to RNA levels in human cases.

2) The studies related to Surfen should be improved. Figure 7 does not adequately demonstrate response of the fish tumour cells to Surfen and the experimental design does not consider transient versus sustained response. As a suggestion to improve these studies, the authors could identify tumours in 30-day old injectants, document GFP fluorescence with Image J or similar and soak the fish just below the MTD of surfen and document the response of individual tumours by loss of GFP fluorescence over time compared to a series of control injectants soaked in vehicle. These experiments should also consider other models (such as PDXs) as well as normal cells.

3) The role of Ras-MAPK-ERK should be explored to a greater extent. Experiments may include a pathway analysis in the zebrafish tumours, as well as the use of a MEK inhibitor to determine whether tumour formation is mitigated.

4) The value of the curved fin model (above measuring tumour formation directly) should be better defined.

5) Experimental detail regarding the zebrafish studies should be added throughout the manuscript. This includes details such as the number of animals used as well as whether single or pooled tumours were used.

*Reviewer #1 (Recommendations for the authors):*

The current manuscript describes a model of Ewing sarcoma wherein the cre-inducible expression of human EWSR1-FLI1 in wild type zebrafish causes the rapid onset of small round blue cell tumors (SRBCTs). The tumors appear to resemble human tumors, expressing CD99 as well as elevated ERK1/2 signaling. Proteomics indicated that progression was associated with dysregulated extracellular matrix metabolism in general and heparan sulfate catabolism in particular. Accordingly, targeting heparan sulfate proteoglycans with Surfen reduced ERK1/2 signaling and tumour cell growth in vitro and in the zebrafish model. Overall, this study reveals a model that may be used to better understand the evolution of sarcoma. However, greater comparisons may be needed in order for this model to be used as a model of human disease.

This model represents an improvement upon previous zebrafish sarcoma models and the data suggest that the methods employed yield tumors that resemble human disease.

The comparison to human disease, and perhaps PDX models, should be enhanced. For example, CD99 staining should be improved and protein levels in human samples should be compared to the fish model, particularly since mRNA levels may not correspond to proteins.

The Surfen studies should be extended to examine the effects on normal cells as well as PDX models.

*Reviewer #2 (Recommendations for the authors):*

1. The authors should comment why they think the ubi promoter is more successful at generating viable transgenic lines that the cmv or β actin promoters.

2. Timing of onset of tumor development is not clearly described in the text but can be ascertained from Figure 1C. The majority of tumors appear to develop at 30 days with smaller numbers appearing at later time points. Detailing this in the Results section would be helpful.

3. The authors compare mass spec data in zebrafish to human RNASeq data without providing any of the primary data. The identifications of the proteins/gene transcripts need to be properly mapped with accompanying p-values of significance.

4. It is unclear why the authors make the direct connection between the proliferative phenotype seen in their zebrafish model and the Ras-MAPK-ERK pathway. Was there evidence of upregulation of this pathway by the proteomic analysis conducted in Figure 2? Did this pathway come up as part of the GSEA? These primary data are not presented in the manuscript.

5. While evidence of pERK expression is subsequently shown by IHC in zebrafish tumors, whether this is a molecular driver is not evaluated. The authors should consider a therapeutic intervention trial with an ERK inhibitor.

6. Is pERK inhibited in vivo in the zebrafish, as was seen in vitro?

7. Straightening of the curvature of the zebrafish fins found to occur following expression of EWS-FLI1 is a surrogate marker of anti-tumor effect and should be described as such in the Discussion. While this may be a helpful preclinical readout, so too is evidence of inhibition of tumor outgrowths, which is more direct. The authors should highlight the advantage of employing the curvature phenotype over the reduced tumor outgrowth.

8. Are the surfen effects sustained or transient? This is a key factor for consideration of therapeutic benefit. If long term studies have not been conducted/planned, the authors should at minimum comment on this issue in the Discussion.

*Reviewer #3 (Recommendations for the authors):*

1. Figure 1C: Did these fish develop other types of tumors? Or was SRBCT the only tumor type caused by ubi:RSG-EWSR1-FLI1? More than half of the fish develop more than one tumor, suggesting that EWSR1-FLI1 is a very potent oncogene with a short latency that requires relatively few collaborating mutations.

2. Figure 1C: Was an attempt made to breed any of the tumor bearing fish? Presumably if the Cre acted in a developing germ cell, then the germ cell would constitutively express EWSR1-FLI. This might block embryologic development. In the event that EWSR1-FLI appeared to be embryologic lethal, did the authors attempt to inject the ubi:RSG-EWSR1-FLI1 construct without Cre? Presumably they would develop stable lines that transmitted and expressed the red fluorescent ubi:RSG-EWSR1-FLI1 construct. These fish could be bred and Cre RNA injected into the one cell fertilized eggs. This might yield stable lines with the ubi:RSG-EWSR1-FLI1 construct integrated at uniform sites for study of tumor onset after Cre mRNA injection.

3. Figure 2E: Are the Western blots from one tumor each or a mixture of multiple tumors?

4. Figure 2F: The authors performed LC-MS/MS mass spectrometry analysis to obtain a set of 194 proteins commonly upregulated in zebrafish tumors compared to surrounding untransformed tissue, which were converted into 194 gene signatures for GSEA using a human Ewing sarcoma RNA-seq dataset. Why not perform RNA-seq analysis in zebrafish tumors for this GSEA analysis? Or examine protein expression by LC-MS/MS mass spectrometry analysis in human Ewing sarcoma lines after shRNA or CRISPR-I knockdown of EWSR1-FLI. In other words, compare protein levels with protein levels or RNA levels with RNA levels by the GSEA approach?

Alternatively, the authors could compare their findings to published RNA-seq results in human Ewing sarcoma without implying that protein expression always recapitulates RNA expression.

5. Figure 2F: Were the zebrafish tumor cells sorted for GFP expression prior to LC-MS/MS mass spectrometry? How cleanly could the tumor tissue be dissected away from normal tissue? How many samples were used for each analysis? How many replicates?

6. Figure 2F: Did the authors also examine genes that are downregulated in the EWSR1-FLI1 expressing tumors compared to control tissue? The downregulation is particularly interesting as later in the manuscript, the authors stated that "Downregulated proteins comprised 65% and 73% of differentially expressed proteins at 24hpf and 48hpf, respectively, suggesting that EWSR1-FLI1 was acting mostly as a transcriptional repressor rather than an activator (Figure 5B)."

7. Figure 3A: The authors stated that "The timeline demonstrates that GFP driven by the ubi promoter is expressed broadly, including throughout the muscle (Figure 3A, middle panel). However, zebrafish expressing eGFP-2A-EWSR1-FLI under the ubi promoter have a distinct distribution pattern of GFP-positive cells (Figure 3A upper panel), predominantly on the fish dorsum, tail and fins (Figure 3A upper panel)." However, Figure 3B shows clearly that nearly 50% of embryos from injection with ubi:RSG-EWSR1-FLI1 + Cre_RNA died by 24hpf. It's very likely that embryos with broader expression pattern died from the toxicity of the transgenes, hence resulted in the "distinct distribution pattern of GFP-positive cells" after the survival selection. I suggest showing the survival curves in Figure 3B before Figure 3A.

8. Figure 3D: How many samples were used for each data point? How many replicates? Was the RT-PCR performed from whole mount embryos? Or from isolated GFP-enriched tissues? Which stage?

9. Figure 3D: Why didn't the authors perform RNAseq instead of qRT-PCR. Then they would establish EWSR1-FLI levels as well as changes in the expression levels of the other zebrafish genes.

10. Figure 3E: At which developmental stage were the samples taken for the WB? Why is the endogenous FLI not detected in the control cells in first three lanes? Apparently this antibody binds to the zebrafish FLI, based on the western blot in Figure 2E. Why not strip this blot and reprobe with anti-GFP antibody. Or make two identical blots and probe one with anti-FLI and one with anti-GFP. This is how one normally documents the expression of a GFP fusion gene. How many samples were used for each analysis? How many replicates?

11. Figure 4A: What does this panel show? Basically some cells are in mitosis and express ph3 and some cells express EWSR1-FLI. This panel can be omitted.

12. Figure 4C: This panel shows phospho-ERK and pH3 expression in control and EWSR1-FLI outgrowths. And the outgrowths express GFP fused to the EWSR1-FLI protein. It is unclear how this panel can be said to show that "activation of ERK signalling is an early event in EWSR1-FLI driven aberrations." Since ERK signalling is shown in the control, I don't see how this panel supports this conclusion. This panel can be omitted.

13. Figure 5: Compares LC-MS/MS mass spectrometry in EWSR1-FLI injected embryos with control injected embryos and shows the proteins that are upregulated and downregulated. This is the correct comparison and is very informative. Why not perform the LC-MS/MS mass spectrometry on human EWSR1-FLI expressing Ewing sarcoma cells compared to shRNA knockdown for the GSEA in Figure 2F, as was done in this figure and look at proteins that are up and down-regulated?

14. Figure 7A: What are the left panel photos? Transmited light? Autofluorescence? Are these the same fish shown at different time points? Or different fish? If the same fish, why did the GFP-EF1 decrease so much at 48 hours? What do the arrows show? Is there an extra arrow in the "+Surfen, 24h" left panel? The current data presentation in this figure is confusing and insufficient to support the authors' statement that "surfen inhibited outgrowth development in the zebrafish embryo model of Ewing sarcoma." How about pERK and pH3? This figure and figure legend need to be clarified to make the point they are trying to make with this study.

15. Figure 7C: Here the authors invented a fin morphology analysis to evaluate the in vivo effect of surfen. However, this novel analysis is poorly characterized and poorly described. Why does surfen lead to smoother edges of the fins?? There is no evidence that the fin curvature is tightly associated with heparin sulfate inhibition, ERK1/2 signaling or Ewing sarcoma cell growth. Please explain.

16. There are two supplementary videos, which are not cited at all in the manuscript?

---

## [Author Response]

Essential revisions:Three experts with have reviewed this manuscript and have made a number of suggestions which should be addressed. It was felt that the study may be considered for publication in eLife, pending the following essential revisions:1) The comparison of the zebrafish tumours to human disease, should be enhanced. For example, protein levels in human samples should be compared to protein levels in the fish model, particularly since mRNA levels may not correspond to proteins. Alternatively, RNA levels in the fish could be compared to RNA levels in human cases.

This question is addressed in the experiments presented in the updated Figure 3 and Figure 2-figure supplement 1. Previously, we showed that observed small round blue cell tumors were positive for human Ewing sarcoma markers such as EWSR1-FLI1, CD99, PAS, NR0B1 (Figure 2A, B, C, D). To enhance the comparison of zebrafish tumors to human disease we aimed to evaluate the expression of nkx2.2a gene in zebrafish tumors. NKX2.2 is an immunohistochemical marker that has been reported to be sensitive and specific for Ewing sarcoma in human (McCuiston and Bishop, 2018). Because there are no available antibodies for zebrafish nkx2.2a we optimized the RNAscope approach to evaluate RNA expression of nkx2.2a. The advantage of the embryonic zebrafish model is the possibility to apply the whole mount in-situ hybridization to determine the gene expression in whole organism. We generated fish mosaically expressing EWSR1-FLI1 and sorted animals with EWSR1-FLI1- induced tumors at 24hpf. Embryos were fixed in 4% PFA and used for double RNAscope staining for nkx2.2a (529751-C2 RNAscope Probe – Dr-nkx2.2a-C2) and eGFP (538851 RNAscope Probe – EGFP-O4). Normally nkx2.2a is expressed in the spinal cord (Figure 3F, control). We found that consistent with human data the EWSR1-FLI1- expressing cells in tumors were also positive for nkx2.2a (Figure 3F, EWSR1-FLI).

To improve staining for the second Ewing sarcoma marker, CD99 we applied IHC technique on advanced zebrafish tumors presented in distinct locations (Figure 2-figure supplement 1C). Our data showed all tumors located in trunk (Figure 2-figure supplement 1C, tumor 1),, caudal fin (Figure 2-figure supplement 1C, tumor 2) and head (Figure 2-figure supplement 1C, tumor 3) express CD99.

To better characterize zebrafish tumors, we performed the proteomics analysis of zebrafish tumors located on dorsal, pectoral, and caudal fins (Figure 2F). We agree with the reviewer, that mRNA levels not always correspond to protein levels. However, it was shown that differentially expressed mRNAs correlate significantly with their protein products (Koussounadis et al., 2015). Based on that research, we identified a list of differentially expressed proteins in all three tumors and compared them to human tumor microarray data (Figure 2F). Our data show significant enrichment of zebrafish differentially expressed proteins in human Ewing sarcoma microarray dataset (NES: 1.501, FWER:0.029), but not in the rhabdomyosarcoma dataset (-1.476, FWER:0.024). To further address the reviewer’s point of comparing protein to protein levels we used a publicly available list of proteins significantly upregulated in human mesenchymal cells after the expression of EWSR1-FLI1 oncofusion considering that expression of the oncofusion in normal human cells mimics best the transformation process (Tanabe et al., 2018) (Table_oncotarget-09-14428-s009). We used that list of significantly upregulated proteins to run the GSEA to evaluate the enrichment of those hits in our zebrafish tumor dataset (Figure 2-figure supplement 1B). We found that the list of proteins was significantly upregulated in zebrafish tumors (NES: 1.540, FWER p-Value: 0.048) confirming similarity between zebrafish and human disease (Figure 2-figure supplement 1B).

Taken together, we showed that zebrafish tumors are positive for routinely used human Ewing sarcoma markers such as EWSR1-FLI1, nkx2.2a, CD99, PAS (Figure 2, Figure 2-figure supplement 1). We characterized tumors (Figure 1C, Figure 1-figure supplement 3, Supplemental File 1) and showed that they recapitulate key aspects of human disease (Figure 2, Figure 2-figure supplement 1).

2) The studies related to Surfen should be improved. Figure 7 does not adequately demonstrate response of the fish tumour cells to Surfen and the experimental design does not consider transient versus sustained response. As a suggestion to improve these studies, the authors could identify tumours in 30-day old injectants, document GFP fluorescence with Image J or similar and soak the fish just below the MTD of surfen and document the response of individual tumours by loss of GFP fluorescence over time compared to a series of control injectants soaked in vehicle. These experiments should also consider other models (such as PDXs) as well as normal cells.

We thank the reviewer for this suggestion. We have performed the proposed experiment(updated Figure 7B, Figure 7-figure supplement 3 A, B). We generated fish mosaically expressing EWSR1-FLI1 and sorted animals with EWSR1-FLI1- induced tumors 2 months later. We selected animals with eGFP-positive tumors and split them into two balanced groups for treatment with surfen and DMSO. Each group had tumors similar in size and location (Figure 7-figure supplement 3 A, B). Prior the treatment we documented the eGFP fluorescence in all tumors. The scheme of treatment of zebrafish tumors included 6 days with 0.2uM surfen, 3 days with 0.8uM surfen, 3 days drug holiday with no treatment, and 3 days of with 0.8uM surfen. The control group was treated with DMSO. The total duration of the experiment was 15 days. We found that surfen significantly reduced the tumor size by 30% during the first 15 days of experiment while tumors treated with DMSO showed increase in tumor size by 46% (updated Figure 7B, Figure 7-figure supplement 3 A, B). Based on this result we conclude that surfen is effective both preventing the outgrowth formation in the embryonic model of Ewing sarcoma and tumor progression in older fish.

3) The role of Ras-MAPK-ERK should be explored to a greater extent. Experiments may include a pathway analysis in the zebrafish tumours, as well as the use of a MEK inhibitor to determine whether tumour formation is mitigated.

To address this comment, we performed ERK1/ERK2 pathway analysis in zebrafish tumors. We used gene set enrichment analysis to determine the enrichment of genes associated with ERK1/2 pathway (GOBP_ERK1_AND_ERK2_CASCADE) in zebrafish tumors. We found that the gene set was significantly enriched in zebrafish tumor proteomic dataset (NES: 1.3867193, FWER p-Value: 0.03) (Updated Figure 4D). The data were also confirmed by IHC staining and western blot analysis for phosphorylated ERK1/2 (Figure 4E, F).

To enhance the experiment, we determined whether surfen and MEK inhibitor trametinib have a comparable effect on outgrowth formation (Figure 7-figure supplement 1). To determine the experimental concentration of trametinib, we treated embryos at 24hpf with 1uM,10uM, 20uM of trametinib, as well as 0.2uM, 1uM surfen or DMSO for 96 hours. We found that survival rate of embryos treated with 1 and 10 μm of trametinib was not significantly different from survival rate of embryos treated with 0.2 μm surfen or DMSO (Figure 7-figure supplement 1A). Next, we looked whether treatment of embryos with 1uM trametinib has effect on ERK phosphorylation. We treated WT embryos at 24hpf with 1uM trametinib, 0.2 μm surfen or DMSO for 24hours. Embryos were used for WB analysis. We found that trametinib strongly decreased the level of ERK1/2 phosphorylation (Figure 7-figure supplement 1B). In contrast, surfen did not affect the total level of ERK1/2 phosphorylation in normal fish (Figure 7-figure supplement 1B). To look specifically at cells expressing EWSR1-FLI1 we generated fish mosaically expressing EWSR1-FLI1, sorted embryos at 24hpf based on eGFP expression, and treated them with 0.2uM surfen, 1uM trametinib or DMSO. After 24h of treatment we fixed embryos in 4%PFA and performed immunofluorescent staining for eGFP and pERK1/2. Consistent with our previous data, we found that cells expressing EWSR1-FLI1 were positive for pERK1/2 (Figure 7-figure supplement 1C, DMSO), however treatment of embryos with both 0.2uM surfen or 1uM trametinib decreased ERK1/2 phosphorylation in EWSR1-FLI positive cells (Figure 7-figure supplement 1C).

Finally, to compare the effect of MEK inhibitor trametinib to surfen on outgrowth formation we generated fish mosaically expressing EWSR1-FLI1 and sorted them based on the presence of eGFP positive outgrowth at 24hpf. Prior the treatment we documented the eGFP fluorescence in all outgrowths. We treated fish with outgrowth with 0.2uM surfen, 1uM trametinib or DMSO for 24h and analyzed the size of outgrowth (Figure 7-figure supplement 1D). We found that treatment of fish with both 1uM trametinib and 0.2 uM surfen resulted in significant decrease in outgrowth size. (Figure 7-figure supplement 1D).

4) The value of the curved fin model (above measuring tumour formation directly) should be better defined.

We have added text to further explain the utility of the assay.

5) Experimental detail regarding the zebrafish studies should be added throughout the manuscript. This includes details such as the number of animals used as well as whether single or pooled tumours were used.

We provided the required information throughout the manuscript.

Reviewer #1 (Recommendations for the authors):The current manuscript describes a model of Ewing sarcoma wherein the cre-inducible expression of human EWSR1-FLI1 in wild type zebrafish causes the rapid onset of small round blue cell tumors (SRBCTs). The tumors appear to resemble human tumors, expressing CD99 as well as elevated ERK1/2 signaling. Proteomics indicated that progression was associated with dysregulated extracellular matrix metabolism in general and heparan sulfate catabolism in particular. Accordingly, targeting heparan sulfate proteoglycans with Surfen reduced ERK1/2 signaling and tumour cell growth in vitro and in the zebrafish model. Overall, this study reveals a model that may be used to better understand the evolution of sarcoma. However, greater comparisons may be needed in order for this model to be used as a model of human disease.This model represents an improvement upon previous zebrafish sarcoma models and the data suggest that the methods employed yield tumors that resemble human disease.The comparison to human disease, and perhaps PDX models, should be enhanced. For example, CD99 staining should be improved and protein levels in human samples should be compared to the fish model, particularly since mRNA levels may not correspond to proteins.The Surfen studies should be extended to examine the effects on normal cells as well as PDX models.

We are grateful to Reviewer 1 for their comments.

This question is addressed in the experiments presented in the updated Figure 3 and Figure2-figure supplement 1. Previously, we showed that observed small round blue cell tumors were positive for human Ewing sarcoma markers such as EWSR1-FLI1, CD99, PAS, NR0B1 (Figure 2A, B, C, D). To enhance the comparison of zebrafish tumors to human disease we aimed to evaluate the expression of nkx2.2a gene in zebrafish tumors. NKX2.2 is an immunohistochemical marker that has been reported to be sensitive and specific for Ewing sarcoma in human (McCuiston and Bishop, 2018). Because there are no available antibodies for zebrafish nkx2.2a we optimized the RNAscope approach to evaluate RNA expression of nkx2.2a. The advantage of the embryonic zebrafish model is the possibility to apply the whole mount in-situ hybridization to determine the gene expression in whole organism. We generated fish mosaically expressing EWSR1-FLI1 and sorted animals with EWSR1-FLI1- induced tumors at 24hpf. Embryos were fixed in 4% PFA and used for double RNAscope staining for nkx2.2a (529751-C2 RNAscope Probe – Dr-nkx2.2a-C2) and eGFP (538851 RNAscope Probe – EGFP-O4). Normally nkx2.2a is expressed in the spinal cord (Figure 3F, control). We found that consistent with human data the EWSR1-FLI1- expressing cells in tumors were also positive for nkx2.2a (Figure 3F, EWSR1-FLI).

To improve staining for the second Ewing sarcoma marker, CD99 we applied IHC technique on advanced zebrafish tumors presented in distinct locations (Figure 2-figure supplement 1C). Our data showed all tumors located in trunk trunk (Figure 2-figure supplement 1C, tumor 1), caudal fin (Figure 2-figure supplement 1C, tumor 2) and head (Figure 2-figure supplement 1C, tumor 3) express CD99.

To better characterize zebrafish tumors, we performed the proteomics analysis of zebrafish tumors located on dorsal, pectoral, and caudal fins (Figure 2F). We agree with the reviewer, that mRNA levels not always correspond to protein levels. However, it was shown that differentially expressed mRNAs correlate significantly with their protein products (Koussounadis et al., 2015). Based on that research, we identified a list of differentially expressed proteins in all three tumors and compared them to human tumor microarray data (Figure 2 F). Our data show significant enrichment of zebrafish differentially expressed proteins in human Ewing sarcoma microarray dataset (NES: 1.501, FWER:0.029), but not in the rhabdomyosarcoma dataset (-1.476, FWER:0.024). To further address the reviewer’s point of comparing protein to protein levels we used a publicly available list of proteins significantly upregulated in human mesenchymal cells after the expression of EWSR1-FLI1 oncofusion considering that expression of the oncofusion in normal human cells mimics best the transformation process (Tanabe et al., 2018) (Table_oncotarget-09-14428-s009). We used that list of significantly upregulated proteins to run the GSEA to evaluate the enrichment of those hits in our zebrafish tumor dataset (Figure 2-figure supplement 1B). We found that the list of proteins was significantly upregulated in zebrafish tumors (NES: 1.540, FWER p-Value: 0.048) confirming similarity between zebrafish and human disease (Figure 2-figure supplement 1B).

Taken together, we showed that zebrafish tumors are positive for routinely used human Ewing sarcoma markers such as EWSR1-FLI1, nkx2.2a, CD99, PAS (Figure 2, Figure 2-figure supplement 1). We characterized tumors (Figure 1C, Figure 1-figure supplement 3, Supplemental File 1) and showed that they recapitulate key aspects of human disease (Figure 2, Figure 2-figure supplement 1).

To see how surfen affects normal and tumor cells we took advantage of our genetic model of Ewing sarcoma and performed treatment of fish with tumors with surfen. We selected animals with eGFP positive EWSR1-FLI1-induced tumors and split them into two balanced groups for treatment with surfen and DMSO (Figure 7-figure supplement 3 A, B). Prior the treatment we documented the eGFP fluorescence in all tumors. We found that adult fish could tolerate higher concentrations of surfen (0.8uM) compared to 0.2uM surfen for embryos. Surfen significantly reduced the tumor size by 30% during the first 15 days of experiment not affecting normal tissue. Tumors treated with DMSO showed increase in tumor size by 46% (updated Figure 7B, Figure 7-figure supplement 3 A, B). Based on that we conclude that surfen is effective preventing the tumor progression in fish demonstrating low toxicity for normal tissue.

Reviewer #2 (Recommendations for the authors):1. The authors should comment why they think the ubi promoter is more successful at generating viable transgenic lines that the cmv or β actin promoters.

We have addressed this comment in the Discussion section.

2. Timing of onset of tumor development is not clearly described in the text but can be ascertained from Figure 1C. The majority of tumors appear to develop at 30 days with smaller numbers appearing at later time points. Detailing this in the Results section would be helpful.

Required information was added in the text.

3. The authors compare mass spec data in zebrafish to human RNASeq data without providing any of the primary data. The identifications of the proteins/gene transcripts need to be properly mapped with accompanying p-values of significance.

We provided required data as a new Supplemental File 2.

4. It is unclear why the authors make the direct connection between the proliferative phenotype seen in their zebrafish model and the Ras-MAPK-ERK pathway. Was there evidence of upregulation of this pathway by the proteomic analysis conducted in Figure 2? Did this pathway come up as part of the GSEA? These primary data are not presented in the manuscript.

To address this comment, we performed ERK1/ERK2 pathway analysis in zebrafish tumors. We used gene set enrichment analysis to determine the enrichment of genes associated with ERK1/2 pathway (GOBP_ERK1_AND_ERK2_CASCADE) in zebrafish tumors. We found that the gene set was significantly enriched in zebrafish tumor proteomic dataset (NES: 1.3867193, FWER p-Value: 0.03) (Updated Figure 4D). The data were also confirmed by IHC staining and western blot analysis for phosphorylated ERK1/2 (Figure 4E, F).

To enhance the experiment, we determined whether surfen and MEK inhibitor trametinib have comparable effect on outgrowth formation (Figure 7-figure supplement 1). To determine the experimental concentration of trametinib, we treated embryos at 24hpf with 1uM,10uM, 20uM of trametinib, as well as 0.2uM, 1uM surfen or DMSO for 96 hours. We found that survival rate of embryos treated with 1 and 10 μm of trametinib was not significantly different from survival rate of embryos treated with 0.2 μm surfen or DMSO (Figure 7-figure supplement 1A). Next, we looked whether treatment of embryos with 1uM trametinib has effect on ERK phosphorylation. We treated WT embryos at 24hpf with 1uM trametinib, 0.2 μm surfen or DMSO for 24hours. Embryos were used for WB analysis. We found that trametinib strongly decreased the level of ERK1/2 phosphorylation (Figure 7-figure supplement 1B). In contrast, surfen did not affect the total level of ERK1/2 phosphorylation in normal fish (Figure 7-figure supplement 1B). To look specifically at cells expressing EWSR1-FLI1 we generated fish mosaically expressing EWSR1-FLI1, sorted embryos at 24hpf based on eGFP expression, and treated them with 0.2uM surfen, 1uM trametinib or DMSO. After 24h of treatment we fixed embryos in 4%PFA and performed immunofluorescent staining for eGFP and pERK1/2. Consistent with our previous data, we found that cells expressing EWSR1-FLI1 were positive for pERK1/2 (Figure 7-figure supplement 1C, DMSO), however treatment of embryos with both 0.2uM surfen or 1uM trametinib decreased ERK1/2 phosphorylation in EWSR1-FLI positive cells (Figure 7-figure supplement 1C).

Finally, to compare the effect of MEK inhibitor trametinib to surfen on outgrowth formation we generated fish mosaically expressing EWSR1-FLI1 and sorted them based on the presence of eGFP positive outgrowth at 24hpf. Prior the treatment we documented the eGFP fluorescence in all outgrowths. We treated fish with outgrowth with 0.2uM surfen, 1uM trametinib or DMSO for 24h and analyzed the size of outgrowth (Figure 7-figure supplement 1D). We found that treatment of fish with both 1uM trametinib and 0.2uM surfen resulted in significant decrease in outgrowth size. (Figure 7-figure supplement 1D).

5. While evidence of pERK expression is subsequently shown by IHC in zebrafish tumors, whether this is a molecular driver is not evaluated. The authors should consider a therapeutic intervention trial with an ERK inhibitor.

Required data can be found in Figure 7-figure supplement 1. (see section ‘Surfen inhibits EWSR1-FLI1 mediated growth in zebrafish model’.)

6. Is pERK inhibited in vivo in the zebrafish, as was seen in vitro?

Yes, that experiment is addressed in (Figure 7-figure supplement 1C).

7. Straightening of the curvature of the zebrafish fins found to occur following expression of EWS-FLI1 is a surrogate marker of anti-tumor effect and should be described as such in the Discussion. While this may be a helpful preclinical readout, so too is evidence of inhibition of tumor outgrowths, which is more direct. The authors should highlight the advantage of employing the curvature phenotype over the reduced tumor outgrowth.

We have addressed this point in the Discussion section.

8. Are the surfen effects sustained or transient? This is a key factor for consideration of therapeutic benefit. If long term studies have not been conducted/planned, the authors should at minimum comment on this issue in the Discussion.

We have performed the proposed experiment (updated Figure 7B, Figure 7-figure supplement 3 A, B). We generated fish mosaically expressing EWSR1-FLI1 and sorted animals with EWSR1-FLI1- induced tumors 2 months later. We selected animals with eGFP positive tumors and split them into two balanced groups for treatment with surfen and DMSO. Each group had tumors similar in size and location (Figure 7-figure supplement 3 A, B). Prior the treatment we documented the eGFP fluorescence in all tumors. The scheme of treatment of zebrafish tumors included 6 days with 0.2uM surfen, 3 days with 0.8uM surfen, 3 days drug holiday with no treatment, and 3 days of with 0.8uM surfen. The control group was treated with DMSO. The total duration of the experiment was 15 days. We found that surfen significantly reduced the tumor size by 30% during the first 15 days of experiment while tumors treated with DMSO showed increase in tumor size by 46% (updated Figure 7B, Figure 7-figure supplement 3 A, B). Based on this result we conclude that surfen is effective both preventing the outgrowth formation in the embryonic model of Ewing sarcoma and tumor progression in older fish.

Reviewer #3 (Recommendations for the authors):1. Figure 1C: Did these fish develop other types of tumors? Or was SRBCT the only tumor type caused by ubi:RSG-EWSR1-FLI1? More than half of the fish develop more than one tumor, suggesting that EWSR1-FLI1 is a very potent oncogene with a short latency that requires relatively few collaborating mutations.

We agree with that comment. In our new Cre-inducible mosaic model of Ewing sarcoma 34% of fish developed SRBCT tumors. Only 1 fish out of 77 developed a leukemia-like SRBCT (see. Section ‘Cre-inducible expression of EWSR1-FLI1 drives SRBCT development in zebrafish’).

2. Figure 1C: Was an attempt made to breed any of the tumor bearing fish? Presumably if the Cre acted in a developing germ cell, then the germ cell would constitutively express EWSR1-FLI. This might block embryologic development. In the event that EWSR1-FLI appeared to be embryologic lethal, did the authors attempt to inject the ubi:RSG-EWSR1-FLI1 construct without Cre? Presumably they would develop stable lines that transmitted and expressed the red fluorescent ubi:RSG-EWSR1-FLI1 construct. These fish could be bred and Cre RNA injected into the one cell fertilized eggs. This might yield stable lines with the ubi:RSG-EWSR1-FLI1 construct integrated at uniform sites for study of tumor onset after Cre mRNA injection.

We have created a stable line expressing Ubi:RSG-EF1 and used it for tissue-specific expression of the oncofusion. That research will be published as a separate paper.

3. Figure 2E: Are the Western blots from one tumor each or a mixture of multiple tumors?

That is a western blot from one tumor. We specified that in the text.

4. Figure 2F: The authors performed LC-MS/MS mass spectrometry analysis to obtain a set of 194 proteins commonly upregulated in zebrafish tumors compared to surrounding untransformed tissue, which were converted into 194 gene signatures for GSEA using a human Ewing sarcoma RNA-seq dataset. Why not perform RNA-seq analysis in zebrafish tumors for this GSEA analysis? Or examine protein expression by LC-MS/MS mass spectrometry analysis in human Ewing sarcoma lines after shRNA or CRISPR-I knockdown of EWSR1-FLI. In other words, compare protein levels with protein levels or RNA levels with RNA levels by the GSEA approach?

To address that comment we used publicly available list of proteins significantly upregulated in human mesenchymal cells after the expression of EWSR1-FLI1 oncofusion considering that expression of the oncofusion in normal human cells mimics best the transformation process (Tanabe at al 2018, oncotarget-09-14428-s009). We used that list of differentially upregulated proteins to run the GSEA to evaluate the enrichment of those hits in our zebrafish tumors dataset (Figure 2-figure supplement 1B). We found that the list of proteins was significantly upregulated in zebrafish tumors (NES: 1.54, FWER p-Value: 0.048) confirming similarity between zebrafish and human disease (Figure 2-figure supplement 1B).

Alternatively, the authors could compare their findings to published RNA-seq results in human Ewing sarcoma without implying that protein expression always recapitulates RNA expression.

We agree with that comment. It was shown that differentially expressed mRNAs correlate significantly with their protein products (Antonis Koussounadis 2015). Based on that research, we identified a list of differentially expressed proteins in all three tumors and compared them to human tumor microarray data (Figure 2F). Our data show significant enrichment of zebrafish differentially expressed proteins in human Ewing sarcoma microarray dataset (NES 1.501, FWER:0.029), but not rhabdomyosarcoma dataset (NES:-1.476, FWER:0.024).

5. Figure 2F: Were the zebrafish tumor cells sorted for GFP expression prior to LC-MS/MS mass spectrometry? How cleanly could the tumor tissue be dissected away from normal tissue? How many samples were used for each analysis? How many replicates?

We did not sort the tumor cells based on GFP expression assuming that could affect stability of proteins involved in ECM. We expected to have minor contamination with normal cells. We performed the analysis of 3 different tumors and 3 normal tissues.

6. Figure 2F: Did the authors also examine genes that are downregulated in the EWSR1-FLI1 expressing tumors compared to control tissue? The downregulation is particularly interesting as later in the manuscript, the authors stated that "Downregulated proteins comprised 65% and 73% of differentially expressed proteins at 24hpf and 48hpf, respectively, suggesting that EWSR1-FLI1 was acting mostly as a transcriptional repressor rather than an activator (Figure 5B)."

We agree that repressed genes are particularly interesting in that research. We found that downregulated proteins were involved in several types of metabolism. We are going to focus on that finding in future research.

7. Figure 3A: The authors stated that "The timeline demonstrates that GFP driven by the ubi promoter is expressed broadly, including throughout the muscle (Figure 3A, middle panel). However, zebrafish expressing eGFP-2A-EWSR1-FLI under the ubi promoter have a distinct distribution pattern of GFP-positive cells (Figure 3A upper panel), predominantly on the fish dorsum, tail and fins (Figure 3A upper panel)." However, Figure 3B shows clearly that nearly 50% of embryos from injection with ubi:RSG-EWSR1-FLI1 + Cre_RNA died by 24hpf. It's very likely that embryos with broader expression pattern died from the toxicity of the transgenes, hence resulted in the "distinct distribution pattern of GFP-positive cells" after the survival selection. I suggest showing the survival curves in Figure 3B before Figure 3A.

We have changed the order of figures.

8. Figure 3D: How many samples were used for each data point? How many replicates? Was the RT-PCR performed from whole mount embryos? Or from isolated GFP-enriched tissues? Which stage?

To characterize the effects of *EWSR1-FLI1* expression during early stages of zebrafish development we integrated the *ubi:RSG-EWSR1-FLI1* cassette into the zebrafish genome in the presence of *Cre* mRNA. Negative controls included co-injection of *Cre* mRNA with an *ubi:RSG* construct that lacks *EWSR1-FLI1* and co-injection of *ubi:RSG-EWSR1-FLI* with *GFP* mRNA. We collected embryos at 24hpf expressing low (GFP+), medium (GFP++) and high (GFP+++) levels of *EWSR1-FLI1* (Figure 3C) for analysis via qRT-PCR. Each group consisted of 10 embryos for RNA extraction and subsequent RT-PCR. The experiment was made in 3 repeats. Each repeat had 3 replicas (see methods section for details).

9. Figure 3D: Why didn't the authors perform RNAseq instead of qRT-PCR. Then they would establish EWSR1-FLI levels as well as changes in the expression levels of the other zebrafish genes.

We assumed that any RNA-seq data should be confirmed by Q-PCR to determine the relative level of RNA expression. Because we were interested in a specific gene expression we applied Q-PCR as a better quantitative alternative.

10. Figure 3E: At which developmental stage were the samples taken for the WB? Why is the endogenous FLI not detected in the control cells in first three lanes? Apparently this antibody binds to the zebrafish FLI, based on the western blot in Figure 2E. Why not strip this blot and reprobe with anti-GFP antibody. Or make two identical blots and probe one with anti-FLI and one with anti-GFP. This is how one normally documents the expression of a GFP fusion gene. How many samples were used for each analysis? How many replicates?

Embryos for western blot were taken at 24hpf. Each sample had 10 embryos. We agree that antibody could recognize the endogenous FLI1 which has different size and could be easily distinguished from EWSR1-FLI1 as a separate band on the gel. We also confirmed that result by Q-PCR using the set of primers recognizing specifically human EWSR1-FLI1. We did not perform western-blot for eGFP because eGFP-2A-EF1 does not represent a fusion protein, but two separate proteins. The western blot analysis was performed at least 3 times for each experiment in 3 replicates (see methods for details).

11. Figure 4A: What does this panel show? Basically some cells are in mitosis and express ph3 and some cells express EWSR1-FLI. This panel can be omitted.

Phospho-H3 is one of the known markers of proliferation (I˙lhan Elmaci, 2017). We have applied IF staining for pH3 to show that EWSR1-FLI1 expression correlates with increased cell proliferation rate.

12. Figure 4C: This panel shows phospho-ERK and pH3 expression in control and EWSR1-FLI outgrowths. And the outgrowths express GFP fused to the EWSR1-FLI protein. It is unclear how this panel can be said to show that "activation of ERK signalling is an early event in EWSR1-FLI driven aberrations." Since ERK signalling is shown in the control, I don't see how this panel supports this conclusion. This panel can be omitted.

We do not agree with the reviewer’s comment. It is an important finding to show that EWSR1-FLI1 expression is associated with activation of ERK1/2 signaling in both embryonic model and advanced zebrafish tumors. ERK1/2 is an important regulator of cell proliferation driving signaling from cell surface receptors. The figure demonstrates that ERK is activated in tumor outgrowths specifically associated with expression of EWSR1-FLI1. There is of course some natural ERK signaling in other parts of the embryo, including the control, for example in what are likely blood cells in the trunk. However, we do not observe ERK activation in the fin in control embryos.

13. Figure 5: Compares LC-MS/MS mass spectrometry in EWSR1-FLI injected embryos with control injected embryos and shows the proteins that are upregulated and downregulated. This is the correct comparison and is very informative. Why not perform the LC-MS/MS mass spectrometry on human EWSR1-FLI expressing Ewing sarcoma cells compared to shRNA knockdown for the GSEA in Figure 2F, as was done in this figure and look at proteins that are up and down-regulated?

We believe that could be an interesting experiment to perform to identify direct EWSR1-FLI effectors in mature tumors. We also used publicly available list of proteins significantly upregulated in human mesenchymal cells after the expression of EWSR1-FLI1 oncofusion considering that expression of the oncofusion in normal human cells mimics best the transformation process (Tanabe at al 2018, oncotarget-09-14428-s009). We used that list of proteins to run the GSEA to evaluate the enrichment of those hits in our zebrafish tumors dataset (Figure 2-figure supplement 1B). We found that list of proteins was significantly upregulated in zebrafish tumors (NES:1.54 FWER p-Value: 0.048) confirming similarity between zebrafish and human disease (Figure 2-figure supplement 1B).

14. Figure 7A: What are the left panel photos? Transmited light? Autofluorescence? Are these the same fish shown at different time points? Or different fish? If the same fish, why did the GFP-EF1 decrease so much at 48 hours? What do the arrows show? Is there an extra arrow in the "+Surfen, 24h" left panel? The current data presentation in this figure is confusing and insufficient to support the authors' statement that "surfen inhibited outgrowth development in the zebrafish embryo model of Ewing sarcoma." How about pERK and pH3? This figure and figure legend need to be clarified to make the point they are trying to make with this study.

Figure 7A represents images of two fish treated with DMSO or surfen for 0, 24 and 48 hours. At the left panel it is a brightfield image to demonstrate the size of the outgrowth, because the eGFP-2A-EWSR1-FLI1 signal is decreasing at later time points which we connect to fish growth and appearance of pigmented melanocytes. The arrows show the outgrowth driven by EWSR1-FLI1. We removed an extra-error. We also performed the quantitative analysis of the outgrowth development after fish treatment with surfen, trametinib and DMSO (Figure 7-figure supplement 1).

To look how surfen and trametinib affect ERK1/2 phosphorylation in EWSR1-FLI1 expressing fish we generated fish mosaically expressing EWSR1-FLI1, sorted embryos at 24hpf based on eGFP expression, and treated them with surfen, trametinib or DMSO. After 24h of treatment we fixed embryos in 4%PFA and performed immunofluorescent staining for eGFP and pERK1/2. Consistent with our previous data, we found that cells expressing EWSR1-FLI1 are positive for pERK1/2 (Figure 7-figure supplement 1C, DMSO), however treatment of embryos with 0.2uM surfen or 1uM trametinib decreased ERK1/2 phosphorylation in EWSR1-FLI positive cells.

15. Figure 7C: Here the authors invented a fin morphology analysis to evaluate the in vivo effect of surfen. However, this novel analysis is poorly characterized and poorly described. Why does surfen lead to smoother edges of the fins?? There is no evidence that the fin curvature is tightly associated with heparin sulfate inhibition, ERK1/2 signaling or Ewing sarcoma cell growth. Please explain.

We have modified the discussion to put this experiment in context.

16. There are two supplementary videos, which are not cited at all in the manuscript?

We added the citations.